# The Relationship between Serum Adipokines, miR-222-3p, miR-103a-3p and Glucose Regulation in Pregnancy and Two to Three Years Post-Delivery in Women with Gestational Diabetes Mellitus Adhering to Mediterranean Diet Recommendations

**DOI:** 10.3390/nu14224712

**Published:** 2022-11-08

**Authors:** Johanna Valerio, Ana Barabash, Nuria Garcia de la Torre, Paz De Miguel, Verónica Melero, Laura del Valle, Inmaculada Moraga, Cristina Familiar, Alejandra Durán, Maria Jose Torrejón, Angel Diaz, Inés Jiménez, Pilar Matia, Miguel Angel Rubio, Alfonso Luis Calle-Pascual

**Affiliations:** 1Endocrinology and Nutrition Department, Hospital Clínico Universitario San Carlos and Instituto de Investigación Sanitaria del Hospital Clínico San Carlos (IdISSC), 28040 Madrid, Spain; 2Centro de Investigación Biomédica en Red de Diabetes y Enfermedades Metabólicas Asociadas (CIBERDEM), 28040 Madrid, Spain; 3Facultad de Medicina, Medicina II Department, Universidad Complutense de Madrid, 28040 Madrid, Spain; 4Clinical Laboratory Department, Hospital Clínico Universitario San Carlos and Instituto de Investigación Sanitaria del Hospital Clínico San Carlos (IdISSC), 28040 Madrid, Spain

**Keywords:** gestational diabetes mellitus, Mediterranean Diet, serum microRNA, adipokines, cytokines, nutritional intervention

## Abstract

The San Carlos Gestational Diabetes Mellitus (GDM) prevention study, a nutritional intervention RCT based on a Mediterranean Diet (MedDiet), has been shown to reduce the incidence of GDM. The objective of this study is to investigate the relationship of leptin, adiponectin, interleukin-6 (IL-6), tumour necrosis factor-alpha (TNF-α), insulin and HOMA-IRand circulating miRNAs (miR-29a-3p, miR-103a-3p, miR-132-3p, miR-222-3p) with the appearance of GDM and with MedDiet-based nutritional intervention, at 24–28 gestational weeks (GW), and in glucose regulation 2–3 years post-delivery (PD). A total of 313 pregnant women, 77 with GDM vs. 236 with normal glucose tolerance (NGT), 141 from the control group (CG, MedDiet restricting the consumption of dietary fat including EVOO and nuts during pregnancy) vs. 172 from the intervention group (IG, MedDiet supplemented with extra virgin olive oil (EVOO) and pistachios during pregnancy) were compared at Visit 1 (8–12 GW), Visit 2 (24–28 GW) and Visit 3 (2–3 years PD). Expression of miRNAs was determined by the Exiqon miRCURY LNA RT-PCR system. Leptin, adiponectin, IL-6 and TNF-α, were measured by Milliplex^®^ immunoassays on Luminex 200 and insulin by RIA. Women with GDM vs. NTG had significantly higher leptin median (Q1–Q3) levels (14.6 (9.2–19.4) vs. 9.6 (6.0–15.1) ng/mL; *p* < 0.05) and insulin levels (11.4 (8.6–16.5) vs. 9.4 (7.0–12.8) µUI/mL; *p* < 0.001) and lower adiponectin (12.9 (9.8–17.2) vs. 17.0 (13.3–22.4) µg/mL; *p* < 0.001) at Visit 2. These findings persisted in Visit 3, with overexpression of miR-222-3p (1.45 (0.76–2.21) vs. 0.99 (0.21–1.70); *p* < 0.05)) and higher levels of Il-6 and TNF-α. When the IG is compared with the CG lower levels of insulin, HOMA-IR-IR, IL-6 levels at Visit 2 and 3 and leptin levels only at Visit 2 were observed. An overexpression of miR-222-3p and miR-103a-3p were also observed in IG at Visit 2 and 3. The miR-222-3p and miR103a-3p expression correlated with insulin levels, HOMA-IR, IL-6 and TNF-α at Visit 2 (all *p* < 0.05). These data support the association of leptin, adiponectin and insulin/HOMA-IR with GDM, as well as the association of insulin/HOMA-IR and IL-6 and miR-222-3p and miR-103a-3p expression with a MedDiet-based nutritional intervention.

## 1. Introduction

Gestational Diabetes Mellitus (GDM) is defined as any degree of glucose intolerance, which develops and/or is first diagnosed during gestation and has been associated with maternal and offspring adverse events and postnatal Type 2 Diabetes Mellitus [1]. The condition is primarily due to the inability of β-cell function to compensate for the insulin resistance (IR) of late pregnancy [2].

There is increasing interest in understanding the pathophysiology of GDM to find biochemical markers with potential diagnostic and therapeutic utility [3,4]. One of the characteristics of GDM is the presence of a low-grade inflammatory state [5] that can be reduced after adopting healthy dietary patterns early in pregnancy [6]. In fact, our group published that adherence to a Mediterranean Diet (MedDiet) supplemented with extra virgin olive oil (EVOO) and pistachios was associated with a 30% reduction in the incidence of GDM [7,8]. Experimental studies have shown that components of the MedDiet may positively modulate the insulin signalling pathway, reducing inflammatory cytokines and adipokines and modifying some microRNA (miRNA) profiles [9].

Both adiponectin and leptin play a role in glucose regulation in pregnancy. Indeed, it has been observed that leptin levels increase significantly during pregnancy in GDM, and these changes are prior to metabolic modifications and changes in adiposity. Conversely, low adiponectin levels have been associated with IR, type 2 diabetes mellitus (T2DM) and GDM. In fact, it has been suggested that the presence of low adiponectin levels and/or high leptin levels before gestation or in early pregnancy may be a useful risk marker for GDM and adiposity [10,11,12,13].

The proinflammatory cytokines tumour necrosis factor-alpha (TNF-α) and interleukin-6 (IL-6) are secreted by the placenta during pregnancy. Increased levels of both cytokines have been reported in GDM compared to control women [2,14]. In fact, TNF-α has been proposed as a possible marker for IR during pregnancy, together with the aforementioned adipokines [15].

The miRNAs are small endogenous single-stranded RNAs between 18 and 22 nucleotides (nt) in length [16] that regulate gene expression at a translational level. Although miRNAs are mainly located in the cytosol, they can be exported to the circulation and regulate gene expression at a distance in recipient cells [17]. MiR-29a-3p, miR-132-3p, miR-103a-3p and miR-222-3p are involved in pancreatic β cells function, insulin signalling, liver metabolism and glucose regulation [18,19,20,21,22,23,24,25]. These miRNAs may also play a role in GDM and could be reliable predictive biomarkers in the early development of GDM [23]. Furthermore, a relationship of serum miRNAs with inflammatory markers, such as Il-6 and TNF-α, has been observed, supporting the role of miRNAs in inflammation [26].

The hypothesis of the current study is that circulating levels of the aforementioned miRNAs could be associated with the appearance of GDM and postnatal abnormal glucose regulation (AGR), acting through the modification of adipokines and insulin sensitivity. Nutritional intervention based on the Mediterranean Diet may lead to changes in the circulating levels of these miRNAs and adipokines. This study therefore assesses the adipokines (leptin and adiponectin), inflammatory cytokines (IL-6 and TNF-α) and the serum expression of miRNAs (miR-29a-3p, miR-103a-3p, miR-132-3p and miR-222-3p) at the beginning of gestation (Visit 1, baseline 8–12 GW), at the time of GDM screening (Visit 2: 24–28 GW) and at 2–3 years PD when T2DM is diagnosed (Visit 3).

The objective of this study is to assess whether adipokine circulating miRNA levels differed between women with GDM compared with NGT and whether this pattern may be modified by a nutritional intervention based on the Mediterranean Diet.

## 2. Materials and Methods

### 2.1. Study Design

This is a secondary analysis of the St. Carlos GDM prevention study: a prospective, single-centre, randomized pregnancy cohort study designed for the prevention of GDM through early nutritional intervention based on a MedDiet (ISRCTN84389045). It was conducted from January 2015 to August 2016. For the current study, we included a sub-cohort of women who subsequently attended a follow-up visit at 2–3 years PD between 2017 and 2018.

The study was approved by the Clinical Trials Committee of the Hospital Clínico San Carlos (CI 13/296-E), Madrid, Spain and conducted according to the Declaration of Helsinki. All women signed the informed consent form at the onset of the trial. A detailed description of the ‘St. Carlos GDM prevention study’ has been previously published [8].

### 2.2. Sample Size Calculation

For the sample size estimate for the primary objective of the current study (miRNAs), a mean in the control group between 8–12 GW of 14.93.10^−5^ for miR-132-3p, 14.10.10^−5^ for miR-29a-3p and 3.09.10^−5^ for miR-222-3p is expected [23]. To achieve a relative increase in the mean of at least 20% in each of the miRNAs in the intervention group for a significance level of 5% and a power of 80%, 144 women in each group would be needed. With this sample size, it would be possible to detect a relative increase in mean adiponectin between the two study groups of more than 12% and a decrease in leptin of 20%, for a significance level of 5% and a power of 80%.

### 2.3. Subjects

The 874 pregnant women who were evaluated in the St. Carlos GDM prevention study were invited to participate in a follow-up study, with a duration of 2–3 years PD. Women attended their first medical visit between 8 and 12 gestational weeks (GW) (Visit 1, or baseline). At that time, inclusion criteria were age ≥18 years and fasting blood glucose (FBG) <92 mg/dL. Exclusion criteria were having multiple gestations, intolerance/allergy to nuts or EVOO, or any medical conditions or pharmacological therapy that could compromise the follow-up program. From that point, recruited women were randomly separated into a control group (CG) or the intervention group (IG) according to their ethnic origin, their age, their weight and their parity. Women in IG were encouraged to increase the consumption of EVOO and pistachios, while those in CG were advised to restrict all fat intake. In summary, all the women received instructions and recommendations on the Mediterranean Diet from the beginning of pregnancy prior to GW 12. The women included in the intervention group (IG: MedDiet-based supplemented with EVOO and pistachios) were recommended to increase the consumption of EVOO (>40 mL/day) and pistachios (>35 g daily), for which they were supplied free of charge with a 5 L carafe of EVOO and 1 Kg of pistachios every 2 months until delivery to guarantee reaching the minimum consumption. The women included in the control group (CG: MedDiet based, with limitation in the consumption of EVOO and nuts) received the same nutritional recommendations except to limit the consumption of EVOO (<40 mL/day) and nuts of any sort (<25 g/day). These recommendations were maintained until delivery. After birth, all women received the same nutritional recommendations based on the MedDiet but without restricting the consumption of EVOO or nuts; nor were EVOO or nuts provided free of charge.

Screening for GDM was performed at 24–28 GW, applying the IADPSG criteria (International Association of Diabetes and Pregnancy Study Group) [27] (Visit 2). All women enrolled in the study were followed during pregnancy and were offered inclusion in the follow-up program and evaluated 2–3 years PD (Visit 3). After delivery, all received the same nutritional recommendations based on MedDiet patterns.

A total of 305 women who had initially agreed to participate in the study (175 from CG and 130 from IG) were unable to attend the 2-year follow-up visit, either due to a change of residence, contact difficulties or personal logistic considerations and were therefore excluded from the analysis. A total of 256 women (124/132, CG/IG) had a new pregnancy before the 3rd year of follow-up and were excluded. Finally, a total sub-cohort of 313 women (141 from CG and 172 from IG) attending the 2–3 year follow-up visit with available serum samples obtained at 8–12 GW (baseline), 24–28 GW and 2–3 years PD were selected for this study. Their socio-demographic, clinical and biochemical characteristics at baseline are shown in Table 1.

### 2.4. Data Collection

#### 2.4.1. Demographic Data

The following information was collected at the baseline visit: age at entry, ethnicity, family history of metabolic disorders such as type 2 diabetes mellitus (T2DM), obstetric history (miscarriages and GDM), number of pregnancies, educational level, employment status, smoking habits (registering whether they were currently smoking, or they smoked until they found out they were pregnant), height and gestational age at entry concurring to the first ultrasound. Pregestational body weight (BW) was self-reported, and body mass index (BMI) was calculated as BW (kg)/height^2^ (m).

#### 2.4.2. Clinical Data

Clinical and anthropometric data measured at each visit included weight (measured without shoes and wearing lightweight clothes) and blood pressure measured with an electronic sphygmomanometer with adequate armlet after resting 10 min in a sitting position (Omron 705IT). Waist circumference and bioimpedance were assessed only in the postpartum evaluation at Visit 3. Bioimpedance was performed with the Medical Body Composition Analyser (SECA mCA515/514, GmbH & Co. KG, Hamburg, Germany). This bioimpedance machine calculates with SECA scale parameters such as lean mass and skeletal muscle mass, fat mass, total body water and total energy expenditure.

#### 2.4.3. Lifestyle Assessment

The mother’s lifestyle was registered at each visit. Adherence to a healthy lifestyle (including physical activity and general healthy eating habits) was evaluated with the Diabetes Nutrition and Complication Trial (DNCT) questionnaire to obtain the nutrition score (NS) and physical activity score. The nutrition score ranges from −12 to 12, and the goal is a value >5. The physical activity score ranges from −3 to 3, and the goal is a value >0. Adherence to the MedDiet was assessed with the 14-point Mediterranean Diet adherence screener questionnaire to obtain the MEDAS score [28]. A more detailed description has been previously published [8,29].

#### 2.4.4. Biochemical and miRNA Analysis

##### Sample Collection and Biochemical Measurements

A blood sample was obtained between 08.00 and 09.00 a.m. after an overnight fast of at least 10 h at each visit. The following parameters were determined: HbA1c, standardized by the International Federation of Clinical Chemistry and Laboratory Medicine; serum levels of HDL-cholesterol by the enzymatic immunoinhibiting method in an Olympus 5800 (Beckman-Coulter, Brea, CA, USA) and serum triglycerides with a colorimetric enzymatic method using glycerol phosphate oxidase *p*-aminophenazone (GPO-PAP). Thyroid stimulating hormone (TSH) was measured by 3rd generation sandwich-chemiluminescence immunoassay with magnetic particles using human TSH mouse monoclonal antibodies in a DXI-800^®^ (Beckman–Coulter) (range for non-pregnant adults 0.38–5.33 µIU/mL), and free thyroxin 4 (FT4) was measured by competitive-chemiluminescence immunoassay in 2 steps with paramagnetic particles, in a DXI-800^®^ (Beckman–Coulter) (range for non-pregnant adults is 5.8–16.4 pg/mL). LDL-cholesterol was calculated with the Friedewald formula and the homeostasis assessment model for insulin resistance (HOMA-IR) calculated as glucose (mmol/L) × insulin (µUI/mL)/22.5.

At Visits 2 and 3, a 2 h 75 g OGTT was performed. Fasting glucose, 1 h and 2 h blood samples were drawn and GDM diagnosis was established according to the IADPSGc at Visit 2 (24–28 GW). Fasting glucose and 2 h glucose levels were used to diagnose glucose intolerance according to ADA guidelines at Visit 3 (2–3 year PD). The metabolic syndrome (MetS) was diagnosed in accordance with the harmonized definition using the specific waist circumference (WC) measurements of the Spanish population with 3 or more of the following: waist circumference (cm) ≥89.5, prediabetes (fasting plasma glucose (mg/dL) ≥100 and/or 2 h glucose levels >139 mg/dL and/or HbA1c (%) ≥5.7), systolic blood pressure (mmHg) ≥130/diastolic blood pressure (mmHg) ≥85, HDL (mg/dL) <50 and triglycerides (g/L) ≥150 [30].

An external quality guarantee program of the SEQC (Sociedad Española de Química Clínica) evaluates the quality of the methods monthly.

##### Adipokine and Cytokine Analysis

For the analysis of adipokines, cytokines and miRNA, a blood sample was collected in a serum separator tube. The samples were allowed to clot at room temperature for 10–30 min and were then centrifuged for 10 min at 2500× *g* rpm (1125g). The upper phases of serum were transferred into a new tube for a second 5 min centrifugation at 3000× *g* at RT. The serum surface was carefully transferred without touching the pellet to separate sterile, nuclease- and pyrogen-free, 1.5-mL vials that were stored immediately. These serum aliquots remained frozen at −80 °C until being consecutively thawed for analysis of adipokines, cytokines or miRNAs.

Serum leptin, IL-6 and TNF-α levels were measured simultaneously using the Multiplex ELISA technique with magnetic beads (HADK2MAG-61K, Milliplex Map, Millipore CA) and reading fluorescence in a Luminex 200 system (Luminex corp. Austin TX, USA). The assays were performed based on the manufacturer’s protocol. To determine adiponectin serum values, samples were diluted at 1:500. Adiponectin concentration was measured with a commercially available radioimmunoassay (RIA) Kit (Millipore Corporation, Linco Research, Inc., St. Charles, MO, USA; HADP-61HK). The analytical sensitivity of the method was 0.78 μg/mL, and the intra-assay and inter-assay variations were 1.8–6.2% and 6.9–9.3%, respectively. Serum insulin concentrations were also assayed by RIA (Diasorin^®^), with a range of 10–200 μU/mL, a sensitivity of 3 μU/mL and an intra- and inter-assay coefficient of variation of 5.5% and 9.7%, respectively.

##### miRNA Isolation and Analysis

Four miRNAs-miR-29a-3p, miR-103a-3p, miR-132-3p and miR-222-3p-were preselected for analysis based on their previously reported association with GDM. For miRNAs extraction, aliquots of stored serum were cold thawed to prevent RNA degradation. A total of 250 μL of serum was transferred to a new collection tube, and it was then centrifuged at 3000× *g* for 5 min. After centrifugation, 200 μL of serum was collected and used for isolation. RNA was isolated using miRNeasy Serum/Plasma Advanced Kit (Qiagen, Hilden, Germany), according to the manufacturer’s protocol, including the optional step of adding into the sample lysis buffer the bacteriophage MS2 RNA as a carrier to improve RNA isolation yield and three RNA spike-in, UniSP2, UniSP4 and UniSP5 as RNA isolation quality control (miRCURY LNA RNA Spike-in kit; cat.339390; Qiagen, Germany). Each sample was finally eluted in 40 µL of nuclease-free water. The RNA purity was determined based on the relationship between the optical density (OD) measured at 260 and 280 nm (OD260/280) and 260 and 230 nm (OD260/230) in a Nanodrop spectrophotometer (Thermo Fisher Scientific). All samples included had values between 1.8 and 2.0. Samples were stored at −80 °C in 40 µL aliquots until use in RT-qPCR.

Reverse transcription was performed by using the miRCURY LNA RT Kit (Cat 339340; Qiagen) following the manufacturer’s protocol, including U6 (spike-in control) as an internal amplification control and a sample of water instead of RNA (mock) as a negative control. Quantitative PCR reactions were performed using the miRCURY LNA SYBR Green PCR Kit (Cat.339347; Qiagen) on a 7500 fast real-time PCR system (Applied Biosystems, USA). All PCRs were performed in triplicate. Two negative controls, water instead of cDNA and the mock sample, which did not generate any signal, were incorporated into the PCR reaction. An interpolate calibrator was also included in each reaction to determine the equal performance of the assay. The presence of hemolysis was analysed by the relationship between the expression of miR-451a and miR-23a-3p. Hemolyzed samples were excluded. The samples were considered positive if the amplification signal occurred at Ct < 37 (before the 37th threshold cycle). All primers were purchased from Qiagen (cat 339306; miRCURY LNA miRNA PCR Assay): miR-222-3p (YP00204551), miR-29a-3p (YP00204698), miR-132-3p (YP00206035), miR-103a-3p (YP00204063), miR-23a-3p (YP00204772), miR-451a (YP02119305), UniSp6 (YP00203954), UniSp5 (YP00203955) UniSp2 (YP00203950) and UniSp4 (YP00203953) (Qiagen, Düsseldorf).

For each miRNA assay, the reaction efficiency of amplification reactions was estimated using the standard curve method analysed using the Pfaffl method [31]. All efficiencies were above 1.99; miR-23a-3p was chosen as the best reference gene for normalization, in agreement with previous publications, as it is known to be relatively stable in serum and plasma and not affected by hemolysis. The relative expression of each miRNA at Visits 2 and 3 was calculated with respect to its expression at Visit 1 using the 2−ΔΔCt method. We considered a fold-change ≥1.5 as the threshold for biologically significant miRNA overexpression.

### 2.5. Statistical Analysis

Continuous variables with normal distribution were expressed as mean ± SD and non-normal variables as median (Q1–Q3). The normality of distribution of continuous variables was tested by the Kolmogorov–Smirnov test. Categorical variables are expressed as number and percentage. All primary analyses were performed on an intention-to-treat basis. Comparison between GDM group and NGT group or IG and CG characteristics for categorical variables were evaluated by the *χ*2 test. For continuous variables, measures were compared with Student’s t test or the Mann–Whitney *U* test if distribution of quantitative variables was or was not normal, respectively. Differences in expression levels of miRNA, cytokine and adipokine circulating levels were evaluated by the Mann–Whitney U test. Since the distribution of miRNA expression levels in each group did not follow a normal distribution, the expression level was log-transformed, and measures were compared applying the Mann–Whitney U test. Correlations between levels of the four miRNAs and biochemical data were assessed using the Spearman’s correlation coefficient. The significance level was established at a *p* value < 0.05. SPSS version 27.0 (IBM Corporation, Armonk, NY, USA) and the Jamovi software (version 1.6.23) [32] were used for statistical analysis.

## 3. Results

A total of 77 (24.6%) women developed GDM, while 236 (75.4%) were NGT. Women with GDM had a pre-gestational body weight, BMI, sBP values and TNF-α levels significantly higher than NGT women: median [Q1–Q3,], (3.2 [2.0–4.7] vs. 2.4 [1.8–3.5] pg/mL; *p* < 0.01. Adiponectin levels were lower in the GDM group than in the NGT (15.3 [11.3–24.9] vs. 20.0 [14.8–26.4] µg/mL; *p* = 0.008). No statistically significant differences were found in all other baseline characteristics between the two groups (Table 2).

The nutrition score and MedDiet score improved significantly in the intervention group at Visit 2, both in women with NGT and in GDM, when compared to baseline scores, and were still significantly higher at Visit 3. Similarly, the nutrition score and MEDAS score had also improved significantly in women from the control group at Visit 3 vs. baseline. When women from the intervention group were compared with those in the control group, only significant differences were observed at Visit 2, in both women with GDM and NGT. At the third visit, 2–3 years PD, only the difference in the MedDiet score in the IG remained statistically significant compared to the CG. Changes in nutrition patterns and exercise are shown in Table 3.

As expected, women with GDM had higher levels of plasma glucose, HbA1c and FPI and HOMA-IR than women with NGT in Visit 2 and continued to be higher at Visit 3. Similarly, adiponectin levels were significantly lower in women with GDM compared to women with NGT, while leptin levels were higher at both Visits 2 and 3, while IL-6 and TNF-α levels were only significantly increased at Visit 3.

The miRNAs expression was higher in women with GDM compared with NGT, but only miR-222-3p at 2–3 years PD was 1.5-fold higher in GDM than in NGT (1.45 (0.76–2.21) vs. 0.99 (0.21–1.70); *p* < 0.05). Table 4 displays the clinical and biochemical differences at both 24–28 GW and 2–3 years postpartum between GDM and NGT women.

A stratified analysis in CG vs. IG groups showed, as expected, a significant improvement in fasting and 2 h OGTT glucose levels and HbA1c values in the IG compared to CG at 24–28 GW. IL-6 and insulin levels and HOMA-IR values were significantly lower in the IG at this time and remained lower at 2–3 years PD, while leptin levels were only significant at Visit 2. The miRNAs expression was higher in women from IG compared with CG in miR-222-3p and miR-103a-3p at 24–28 GW and 2–3 years PD. There were no differences in other biochemical parameters at either 24–28 GW or 2–3 years PD (Table 5).

The body composition and each metabolic syndrome component at 2–3 years post-delivery are shown in Table 6.

Women in the GDM group in relation to those in the NGT group presented higher BMI and WC and had significantly higher rates of abnormal glucose regulation (AGR) (25 (36.4%) vs. 28 (19.6%); *p* < 0.001); overweight and obesity (44 (57.1%) vs. 100 (42.5%); *p* < 0.05) and metabolic syndrome (18 (23.4%) vs. 26 (11.0%); *p* < 0.05) at 2–3 years postpartum. However, body composition was similar in both groups.

A correlation analysis was performed to explore a link between the serum expression of miR-222-3p and miR-103a-3p, adiposity, inflammation and insulin resistance. The statistically significant correlations (R Spearman’s) found in crude or BMI-adjusted analysis are presented in Table 7. After correction for pre-gestational BMI, miR-222-3p showed a negative correlation with insulin, TNF-α and IL-6 levels at 24–28 GW. Serum miR-103a-3p expression at 24–28 GW and at 2–3 PD was negatively correlated with insulin, HOMA-IR, leptin and TNF alpha levels at Visit 2. There is a particularly strong negative correlation (*p* < 0.001) between miR-103a-3p expression and IL-6 levels at all three visits. There was also a strong correlation (*p* < 0.001) between both miR-222-3p and miR-103a-3p expression and the change in insulin, HOMA-IR and leptin levels from Visit 1 to Visit 2. Statistically significant correlations (R Spearman’s) of miR-222-3p and miR-103a-3p with variables are displayed in Table 7.

## 4. Discussion

According to the data found in the current study, women who develop GDM have significantly lower adiponectin values at the beginning of gestation along with higher BMI, sBP and TNF-α when compared to women who remain NGT, suggesting a role for adiponectin/TNF-α in GDM risk. However, the role of adiponectin or TNF-α as a predictor of the risk of developing GDM is substantially less important than that of pre-gestational BMI. Nor were any of the other potential biomarkers studied superior to BMI in their association with GDM.

Adiponectin and leptin are secreted by adipose tissue and the placenta, and both have an effect on glucose homeostasis during pregnancy. Previous studies in pregnant women have shown that as pregnancy progressed, serum levels of leptin increased in women with GDM [11,33]. Some studies related GDM to a placental inflammatory component leading to increased leptin production [34,35]. However, to date there have been very few studies focusing on adipokines with follow-up during pregnancy, with scarce data after delivery.

The metabolic changes observed in our study associated with the appearance of GDM (24–28 GW) were an increase in insulin and leptin levels with a simultaneous decrease in adiponectin levels and in insulin sensitivity. These alterations persisted 2–3 years postpartum and were associated with an over-expression of miR-222-3p and higher levels of IL-6 and TNF-α. These results are in line with a previous prospective study which detected a relationship between leptin, adiponectin and insulin resistance that persisted after pregnancy despite unchanged body fat content and distribution. GDM was associated with subclinical inflammation, and affected women showed increased plasma leptin and lower plasma adiponectin levels independently of the degree of insulin sensitivity or obesity [36].

Can these changes be prevented with a nutritional intervention? We have found that the latter, based on the principles of the Mediterranean Diet, when applied early in pregnancy significantly reduced the elevation of the levels of insulin, IL-6, leptin and HOMA-IR values at the time of screening for GDM and 2–3 years post-delivery, as well as the concomitant over-expression of miR-103a-3p and miR-222-3p. These data indicate that a nutritional intervention based on the Mediterranean Diet result in reduced leptin and insulin levels and over-expression of miR-103a-3p and miR-222-3p. These changes are similar both during gestation and postnatally. These results reinforce the hypothesis that leptin may be involved in both the pathway of GDM and the protective effect of the MedDiet on GDM development, irrespective of body weight.

Serum expression of miR-222-3p and miR-103a-3p were both inversely correlated with TNF-α, IL-6, FPI and HOMA-IR values at 24–28 GW and 2–3 years PD. When controlled for pre-gestational BMI, all correlations in the expression of these miRNAs with insulin, HOMA-IR, leptin, TNF-α and IL-6 as well as with its changes in relation to Visit 1 persisted at 24–28 GW and 2–3 years PD.

Few studies have investigated miRNA expression in sera from women with GDM, and their results are highly contradictory. Shi et al. [37] analyzed omental adipose tissue in women with GDM at 38–39 weeks of gestation and observed increased miR-222-3p expression in serum. The authors suggested that miR-222-3p could play an important role in the development of insulin resistance. In fact, in an experimental assay with rats, miR-222-3p was observed to cause downregulation of the insulin receptor substrate-1 (IRS-1) and consequent inactivation of other proteins of the insulin cascade with alteration of glucose transporter 4 (GLUT4) [20]. Ali Tagoma et al. [37] analyzed miRNAs in plasma from 13 pregnant women and observed that out of 84 miRNAs there was an increased expression of miR-195. Some studies have described that miRNAs, such as miR-29a-3p, miR-132-3p and miR-222-3p, show significant differences in their expression in serum in the first weeks of pregnancy [23]. Moreover, Wander et al. [24] observed an increased expression of miR-29a-3p in those women with GDM and male offspring. The accumulated evidence appears to indicate an important role for miR-132-3p and MiR-222-3p expression in glucose homeostasis, regulation of adipose tissue, and GLUT4 [22,37,38]. However, the data obtained in the present study do not suggest any involvement of miR-29a-3p or miR-132-3p with GDM in our sample.

Some experimental and longitudinal studies associate miR-103a-3p with T2DM and cardiovascular diseases. MiR-103a-3p is found in adipose tissue cells, regulating mRNA expression, and causing insulin resistance by altering GLUT4, IRS-1 and caveolin-1 (cav-1) [39]. As mentioned, in our study group we found a positive correlation between miR-103a-3p and high levels of leptin and TNF-α and low levels of adiponectin in women at 24–28 GW that was maintained 2–3 years postpartum. However, in a recent study evaluating the possibility of using miRNAs as a diagnostic test for fetal hypoxia in plasma at 32 and 37 GW, no significant differences in miR-103a-3p were found [40].

The current study shows that women with GDM when compared with women who remained NGT during pregnancy had a higher BMI and waist circumference at 2–3 years postpartum with higher rates of abnormal glucose regulation (36%) and metabolic syndrome (23%). These data indicate that GDM represents a very important risk factor for glucose dysregulation and cardiovascular disease in the postnatal period. These changes are associated with higher levels of insulin, HOMA-IR, leptin and TNF-α than the NGT group, as well as a significantly greater change and increment of these three variables from their baseline levels.

In summary, the data obtained in this study show that women who will develop GDM (24–28 GW) already have significantly lower levels of adiponectin and higher values of BMI and sBP at the beginning of pregnancy (8–12 GW). The metabolic changes associated with the appearance of GDM are a significantly greater increase in insulin and leptin levels and a reduction in adiponectin levels than what is observed in women not developing GDM. These changes induce an increase in insulin resistance. These data are similar to those found in the postnatal assessment of T2DM and are associated at both times with an overexpression of miR-222-3p.

Nutritional intervention based on the principles of the Mediterranean Diet is accompanied by a significant decrease in the levels of insulin, IL-6 and leptin and in HOMA-IR values at the time of the GDM screening (24–28 GW) associated with an overexpression of miR-103a-3p in addition to miR-222-3p. These changes persist at the time of postnatal screening T2DM (2–3 years PD) except for leptin, which loses its statistical significance. Pistachios and EVOO are rich in unsaturated fatty acids, magnesium, and other phytochemical constituents with potential beneficial effects on insulin sensitivity, fasting glucose levels and inflammation. We hypothesize that their antioxidant capacity, given their high levels of lutein, β-carotene and γ-tocopherol, may enhance miR-103a-3p and miR-222-3p overexpression and improve the inflammatory cytokine profiles linked to GDM development.

### Limitations and Strengths

Several limitations were found in our study. First, when MedDiet-based nutritional intervention was assessed, differences in diet between groups may have not been substantial enough to induce changes in sera adipokines and miRNAs. Both groups (IG and CG) received recommendations based on the Mediterranean Diet, reinforced with EVOO and pistachios or restricted consumption of total fats during gestation, and both groups received the same recommendations after delivery. The differences obtained in the score of the questionnaires used between both groups were over two points during pregnancy/after delivery, a difference that can be considered insufficient to detect these effects. On the other hand, differences in the intake of fat-soluble vitamins, which have not been evaluated in the study, could also affect the results. Secondly, the wide methodological ranges in the detected values of adipokines and miRNAs, with SDs values elevated to the point that they can in fact be higher than the calculated means make statistically significant differences more difficult to attain. Another important limitation concerns the location of body fat in body composition. Ectopic fat is more frequent in the presence of low levels of adiponectin. Therefore, our study would probably not have detected significant differences between women with GDM and NGT.

However, in this prospective study, each woman was analysed at three successive times, early in pregnancy (8–12 GW), at the time of screening for GDM (24–28 GW) and at the time of postnatal screening for type 2 diabetes mellitus. This approach permitted analysis of the dynamics of the parameters studied compared to early pregnancy, as well as the differences/similarities of the risk factors involved in GDM and postnatal T2DM. These aspects are the greatest strengths of our study.

## 5. Conclusions

In conclusion, the nutritional intervention based on the principles of the Mediterranean Diet is associated with an overexpression of miR-222 and miR-103 that correlates with an improvement in insulin sensitivity and in the inflammatory profile of cytokines.

Measurement of these parameters can contribute to early detection of abnormal glucose regulation during pregnancy and in the postnatal period. Furthermore, they underline the importance of strategies directed towards nutritional intervention based on the Mediterranean Diet early in pregnancy.

## Figures and Tables

**Table 1 nutrients-14-04712-t001:** Baseline characteristics at 8–12 GW of women analysed.

	Control Group *n* = 141	Intervention Group *n* = 172	*p*
Age (years)	33.6 ± 4.8	34.5 ± 4.7	0.112
Race/Ethnicity			
Caucasian	85 (61.3)	113 (65.7)	0.020
Hispanic	50 (35.5)	56 (32.6)	
Others	6 (4.2)	3 (1.8)	
Family history of			
Type 2 Diabetes	31 (21.9)	49 (28.5)	0.174
MetS (>2 components)	26 (18.4)	42 (24.4)	
Previous history of			
- Gestational DM	7 (5.0)	6 (3.5)	0.791
- Miscarriages	49 (34.7)	61 (35.5)	
Educational status			
Elementary education	15 (10.6)	9 (5.2)	0.109
Secondary School	67 (47.5)	73 (42.4)	
University Degree	57 (40.4)	89 (51.7)	
UNK	2 (1.4)	1 (0.6)	
Employed	105 (74.5)	140 (81.4)	0.510
Number of pregnancies			
Primiparous	45 (31.9)	60 (35.1)	0.214
Second pregnancy	49 (34.8)	73 (42.4)	
>2 pregnancies	47 (33.3)	39 (22.5)	
Smoker			
Never	77 (54.6)	95 (55.2)	0.701
Current	9 (6.4)	14 (8.1)	
Gestational Age (weeks) at baseline	12.1 ± 0.6	12.1 ± 0.5	0.838
Pre-pregnancy self-reported body Weight (kg)	60.0 (53.5–67.0)	58.2 (54.5–65.9)	0.744
Pregnancy Body Weight at entry, 8–12 GW (kg)	61.6 (55.0–69.0)	60.0 (58.8–68.6)	0.910
Weight gain at 8–12 GW	2.0 (0.5–3.8)	1.7 (0.2–3.1)	0.053
Pre-pregnancy BMI (kg/m^2^)	22.6 (20.4–25.5)	22.7 (20.5–25.1)	0.927
Systolic BP/Diastolic BP (mm Hg)	105 ± 9/63 ± 8	107 ± 11/66 ± 9	0.415/0.367
Fasting Blood Glucose (mg/dL)	80.9 ± 6.4	80.8 ± 6.2	0.917
HbA1c (%)	5.1 ± 0.1	5.2 ± 0.2	0.563
Fasting Cholesterol (mg/dL)	175 ± 32	176 ± 27	0.725
Fasting Tryglycerides (mg/dL)	87 ± 39	89 ± 56	0.767
Fasting Plasma Insulin (µUI/mL)	12.3 (8.1–28.1)	11.8 (6.1–21.0)	0.09
HOMA-IR	2.6 (1.6–5.5)	2.3(1.2–4.2)	0.089
Adiponectin (µg/mL)	20.4 (14.1–27.6)	17.7 (13.3–22.5)	0.081
IL-6 (pg/mL)	3.0 (1.5–7.3)	2.0 (0.1–4.9)	0.071
Leptin (ng/mL)	7.3 (4.5–13.2)	6.5 (3.6–10.9)	0.099
TNF-α (pg/mL)	2.9 (2.1–4.4)	2.4 (1.7–3.3)	0.071
TSH mcUI/mL	1.84 ± 1.17	1.98 ± 1.46	0.394
FT4 (pg/dL)	8.68 ± 1.59	8.76 ± 1.35	0.656
MEDAS Score	4.8 ± 1.7	5.1 ± 1.61	0.096
Physical Activity Score	−1.7 ± 1.1	−1.9 ± 1.0	0.251
Nutrition Score	0.9 ± 3.3	0.6 ± 3.2	0.452
GDM at 24–28 GW n (%)	40 (28.4)	37 (21.5)	0.102

Data are given as mean ± SD, median (Q1–Q3) or number (%); control group, MedDiet based, with limitation in the consumption of EVOO and nuts; intervention group, MedDiet-based supplemented with EVOO (>40 mL/day) and pistachios (>35 g/day); MetS: metabolic syndrome; UNK: unknown; BMI: body mass index; GW: gestational weeks; BP: blood pressure; IL-6: interleukin 6; TNFα: tumour necrosis factor-alpha; HOMA-IR, Homeostasis assessment model for insulin resistance; HbA1c, hemoglobin A1c TSH: thyroid stimulating hormone; FT4: free thyroxin 4; MEDAS Score: 14-point Mediterranean Diet adherence screener (MEDAS); physical activity score: (walking daily (>5 days/week) score 0: at least 30 min, score +1, if >60 min, score −1, if <30 min; climbing stairs (floors ⁄ day, >5 days a week): score 0: between 4 and 16: score +1, >16, score −1, <4); GDM, gestational diabetes mellitus after IADPSG criteria. HOMA-IR, Homeostasis assessment model for insulin resistance, HbA1c, hemoglobin A1c.

**Table 2 nutrients-14-04712-t002:** Comparison of baseline (at 8–12 gestational weeks) characteristics of women with gestational diabetes mellitus (GDM) and normal glucose tolerance (NGT) analysed.

	GDM *n* = 77	NGT *n* = 236	*p*
Age (years)	34.9 ± 4.6	33.8 ± 4.8	0.056
Race/Ethnicity			
Caucasian	51 (66.2)	147 (62.3)	0.132
Hispanic	24 (31.2)	82 (34.7)	
Others	2 (2.6)	7 (2.9)	
Family history of			
Type 2 Diabetes	21 (27.3)	59 (25.0)	0.556
MetS (>2 components)	16 (20.8)	52 (22.0)	
Previous history of			
- Gestational DM	4 (5.2)	9 (3.8)	0.513
- Miscarriages	24 (31.2)	87 (36.8)	
Educational status			
Elementary education	8 (10.4)	16 (6.8)	0.303
Secondary School	37 (48.1)	103 (43.7)	
University Degree	31 (40.3)	115 (48.7)	
UNK	1 (1.3)	2 (0.8)	
Employed	64 (83.1)	181 (76.7)	0.453
Number of pregnancies			
Primiparous	29 (37.7)	76 (32.3)	0.760
Second pregnancy	28 (36.4)	94 (40.0)	
>2 pregnancies	20 (25.9)	39 (22.5)	
Smoker			
Never	40 (51.9)	132 (55.9)	0.843
Current	7 (9.1)	16 (6.8)	
Gestational Age (weeks) at baseline	12.2 ± 0.7	12.1 ± 0.5	0.135
Pre-pregnancy Body Weight (kg)	62.0 (55.5–70.5)	58.0 (54.0–65.0)	0.017
Pregnancy Body Weight at entry (kg)	62.0 (56.6–72.5)	60.0 (55.0–67.4)	0.047
Weight gain at 8–12 GW	2.0 (0.0–3.0)	2.0 (0.2–3.5)	0.492
Pre-pregnancy BMI (kg/m^2^)	24.0 (20.8–27.4)	22.2 (20.4–24.8)	0.006
Systolic BP/Diastolic BP (mm Hg)	107 ± 11/66 ± 9	105 ± 9/63 ± 8	0.020/0.314
Fasting Blood Glucose (mg/dL)	81.9 ± 6.2	80.5 ± 6.3	0.084
A1c (%)	5.2 ± 0.2	5.0 ± 0.1	0.540
Fasting Cholesterol (mg/dL)	179 ± 35	174 ± 28	0.329
Fasting Tryglycerides (mg/dL)	96 ± 44	85 ± 50	0.139
Fasting Plasma Insulin (µUI/mL)	11.9 (6.8–24.7)	10.8 (6.6–17.7)	0.322
HOMA-IR	2.5 (1.3–5.0)	2.3 (1.3–3.5)	0.324
Adiponectin (µg/mL)	15.3 (11.3–24.9)	20.0 (14.8–26.4)	0.008
IL-6 (pg/mL)	3.4 (0.9–7.3)	2.2 (0.1–5.7)	0.092
Leptin (ng/mL)	8.8 (5.2–13.2)	6.4 (3.6–11.6)	0.099
TNF-α (pg/mL)	3.2 (2.0–4.7)	2.4 (1.8–3.5)	0.010
TSH µUI/mL	1.84 ± 1.21	1.94 ± 1.36	0.608
FT4 (pg/mL)	8.60 ± 1.39	8.76 ± 1.49	0.471
MEDAS Score	5.1 ± 1.6	5.0 ± 1.7	0.665
Physical Activity Score	−1.9 ± 0.9	−1.8 ± 1.1	0.605
Nutrition Score	0.9 ± 3.0	0.7 ± 3.3	0.517
Intervention Group n (%)	37 (48.1)	135 (57.2)	0.102

Data are mean ± SD, median (Q1-Q3) or number (%); METS: metabolic syndrome; UNK: unknown; BMI: body mass index; GW: gestational weeks; GDM: gestational diabetes mellitus; NGT: normal glucose tolerance; BP: blood pressure; HOMA-IR: insulin resistance index; IL-6: interleukin 6; TNF-α: tumour necrosis factor-alpha; FT4: free thyroxin 4; MEDAS Score, 14-point Mediterranean Diet adherence screener (MEDAS); physical activity score, (walking daily (>5 days/week) score 0, at least 30 min; score +1, if >60 min; score −1, if <30 min; climbing stairs (floors/day, >5 days a week): score 0, between 4 and 16; score +1, >16; score −1: <4); GDM: gestational diabetes mellitus after IADPSG criteria.

**Table 3 nutrients-14-04712-t003:** Lifestyle pattern during gestation and at 2–3 years post-delivery: data by groups and glucose tolerance during gestation.

	Control Group			Intervention Group		
	All	Glucose Regulation		All	Glucose Regulation	
		NGT	GDM		NGT	GDM
N	141	101	40	172	135	37
Pregestational						
Nutrition Score	0.9 ± 3.3	0.7 ± 3.3	1.3 ± 3.4	0.6 ± 3.2	0.6 ± 3.3	0.7 ± 2.6
Phy_Activity_S	−1.7 ± 1.1	−1.7 ± 1.1	−1.8 ± 1.0	−1.9 ± 1.0	−1.9 ± 1.0	−1.9 ± 0.9
MEDAS Score	4.8 ± 1.7	4.8 ± 1.7	4.8 ± 1.8	5.1 ± 1.6	5.1 ±1.7	5.3 ± 1.3
24–28 GW						
Nutrition Score	1.3 ± 3.5	1.0 ± 3.2	2.2 ± 3.9	4.7 ± 3.0 ***^a^	4.8 ± 3.2 ***^a^	4.2 ± 2.4 **^a^
Phy_Activity_S	−1.8 ± 1.0	−1.7 ± 1.0	−1.9 ± 1.0	−1.9 ± 0.9	−1.9 ± 0.9	−1.9 ± 0.8
MEDAS Score	4.7 ± 1.7	4.5 ± 1.6	4.9 ± 2.0	6.7 ± 1.8 ***^a^	6.7 ± 1.8 ***^a^	6.5 ± 1.7 **^a^
2–3 years PD						
Nutrition Score	2.8 ± 3.8 ^b^	2.2 ± 3.9 ^b^	3.8 ± 3.4 ^b^	3.4 ± 3.6 ^b^	3.2 ± 3.8 ^b^	3.9 ± 3.0 ^b^
Phy_Activity_S	−1.6 ± 1.0	−1.6 ± 1.1	−1.7 ± 0.8	−1.7 ± 0.9	−1.7 ± 1.0	−1.9 ± 0.9
MEDAS Score	6.1 ± 1.9 ^b^	5.9 ± 1.9 ^b^	6.5 ± 1.7 ^b^	6.6 ± 1.9 *^a^	6.5 ± 1.9 *^a^	6.9 ± 2.0 ^a^

Data are mean ± SDM or number (%); control group, MedDiet based, with limitation in the consumption of EVOO and nuts; intervention group, MedDiet-based supplemented with EVOO (>40 mL/day) and pistachios (>35 g/day); GW: gestational week; MEDAS score: 14-point Mediterranean Diet adherence screener; Phy_Activity_S: physical activity score; NGT: normal glucose tolerance; GDM: gestational diabetes mellitus; PD: post-delivery; ***, *p* < 0.001, ** *p* < 0.01 and * *p* < 0.05, denote differences between both cohorts; ^a^
*p* < 0.01 and ^b^
*p* < 0.05 denote differences in relation to baseline.

**Table 4 nutrients-14-04712-t004:** Clinical and laboratory data during pregnancy and 2–3 years post-delivery by groups.

	NGT (*n* = 236)		GDM (*n* = 77)	
	24–28 GW	2–3 Years PD	24–28 GW	2–3 Years PD
Body Weight (Kg)	67.4 ± 10.9	63.9 ± 12.9	70.1 ± 11.0	65.4 ± 12.8
FP Glucose (mg/dL)	83.5 ± 5.1	91.9 ± 6.1	92.6 ± 6.7 ***	97.4 ± 7.1 ***
1 h OGTT (mg/dL)	118.8 ± 27.4	n.a.	159.4 ± 33.3 ***	n.a.
2 h OGTT (mg/dL)	104.5 ± 20.8	95.0 ± 19.4	135.1 ± 31.3 ***	108.3 ± 33.4 **
sBP (mm Hg)	104 ± 11	111 ± 12	108 ± 11 *	113 ± 10
dBP (mm Hg)	62 ± 9	71 ± 9	65 ± 8 *	73 ± 8 *
T-Chol. (mg/dL)	255 ± 44	176 ± 29	250 ± 46	183 ± 34
Triglycerides (g/L)	162 ± 55	75 ± 42	168 ± 45	92 ± 55 **
HbA1c-IFCC %	4.9 ± 0.3	5.3 ± 0.3	5.1 ± 0.3 ***	5.5 ± 0.3 **
TSH µUI/mL	2.0 ± 1.0	1.9 ± 1.0	2.1 ± 1.0	1.9 ± 1.0
FT4 (pg/mL)	7.1 ± 1.1	8.3 ± 1.1	6.8 ± 1.0	8.1 ± 1.3
FP insulin (µUI/mL)	9.4 (7.0–12.8)	8.7 (6.7–11.9)	11.4 (8.6–16.5) ***	10.5 (8.1–16.3) **
Increase BS	5.8 ± 15.4	6.1 ± 17.5	7.1 ± 7.0 *	11.1 ± 16.9 *
Change (%)	32 ± 8	28 ± 5	30 ± 3	35 ± 4 *
HOMA-IR	1.8 (1.4–2.7)	1.9 (1.5–2.8)	2.3 (1.3–3.9) *	2.5 (1.9–3.9) *
Increase BL	−0.7 ± 4.5	0.4 ± 5.2	0.1 ± 3.2 *	−1.2 ± 4.4 **
Change (%)	7 ± 4	7 ± 2	8 ± 4	20 ± 11 **
Adiponectin (µg/mL)	17 (13–22)	16.9 (12.0–21.8)	12.9 (9.8–17.2) ***	13.7 (9.5–17.3) **
Increase BL	−2.6 ± 7.1	−3.5 ± 7.2	−3.6 ± 6.8	−3.5 ± 6.9
Change (%)	11 ± 4	13 ± 4	20 ± 3	21 ± 3
IL-6 (pg/mL)	2.1 (0.1–5.1)	2.6 (0.1–6.2)	2.8 (1.3–7.3)	3.4 (1.0–10.5) *
Increase BL	0.3 ± 7.0	−0.1 ± 13.3	0.8 ± 4.5	−0.5 ± 7.5
Change (%)	16 ± 48	53 ± 19	24. ± 21	−42 ± 19
Leptin (ng/mL)	9.6 (6.0–15.1)	6.4 (3.4–11.9)	14.6 (9.2–19.4) *	9.9 (5.6–15.3) **
Increase BL	3.6 ± 5.8	0.2 ± 5.1	5.6 ± 6.4 **	1.5 ± 7.3 *
Change (%)	39 ± 9	2 ± 1	41 ± 7	25 ± 4 *
TNF-α (pg/mL)	2.7 (1.8–3.9)	2.8 (1.9–4.2)	3.0 (2.0–4.4)	3.6 (2.5–4.6) *
Increase BL	0.3 ± 2.2	1.8 ± 1.9	1.9 ± 1.4	1.2 ± 1.6
Change (%)	10 ± 9.1	19 ± 8.5	38 ± 3	31 ± 5
miR-222-3p	1.20 (0.28–1.85)	0.99 (0.21–1.70)	1.24 (0.58–2.01)	1.45 (0.76–2.21) *
miR-103a-3p	0.92 (−0.90–2.24)	1.44 (−0.21–2.53)	1.18 (−0.25–1.91)	1.58 (0.21–2.62)
miR-132	0.18 (−0.35–0.85)	0.05 (−0.55–0.75)	0.18 (−0.59–0.96	0.45 (−0.09–0.94)
miR-29a-3p	0.26 (−0.95–1.67)	0.11 (−1.07–1.54)	0.21 (−0.52–2.06)	0.38 (−0.69–1.81)

Data are mean ± SDM, median IQR or number (%); OGTT: oral glucose tolerance test; sBP: systolic blood pressure; dBP: diastolic blood pressure; HbA1c-IFCC: glycated haemoglobin International Federation of Clinical Chemistry; BL: from baseline; TSH: thyroid-releasing hormone; FT4: free thyroxin 4; FP: fasting plasma insulin; HOMA-IR: homeostasis model assessment of insulin resistance index; GW: gestational weight; PD: post-delivery; WC: waist circumference, BMI: body mass index; IL-6: interleukin 6; TNFα: tumour necrosis factor-alpha;, miR: microRNA; * *p* < 0.05; ** *p* < 0.01 and *** *p* < 0.001 denote differences between both cohorts.

**Table 5 nutrients-14-04712-t005:** Clinical and laboratory during pregnancy and 2–3 years post-delivery data by groups.

	Contral Group (*n* = 141)		Intervention Group (*n* = 172)	
	24–28 GW	2–3 Years PD	24–28 GW	2–3 Years PD
Body Weight (Kg)	68.1 ± 11.8	64.4 ± 14.6	68.1 ± 10.3	64.2 ± 11.2
FP Glucose (mg/dL)	86.3 ± 6.7	93.6 ± 6.9	85.3 ± 6.8 *	92.9 ± 6.7
1 h OGTT (mg/dL)	126.8 ± 31.3	n.a.	124.2 ± 33.5 *	n.a.
2 h OGTT (mg/dL)	110.0 ± 27.0	95.0 ± 23.7	109.6 ± 24.5	99.3 ± 25.7
sBP (mm Hg)	104 ± 11	111 ± 12	105 ± 12	112 ± 12
dBP (mm Hg)	62 ± 9	71 ± 8	63 ± 9	71 ± 9
T-Chol. (mg/dL)	250 ± 47	174 ± 29	257 ± 42	180 ± 31
Triglycerides (g/L)	162 ± 52	83 ± 43	164 ± 54	76 ± 49
HbA1c-IFCC %	5.1 ± 0.3	5.3 ± 0.3	5.0 ± 0.3 *	5.3 ± 0.3
TSH µUI/mL	1.9 ± 1.0	1.9 ± 1.1	2.1 ± 0.9	1.9 ± 0.9
FT4 (pg/mL)	7.2 ± 1.1	8.3 ± 1.1	7.0 ± 1.1	8.1 ± 1.3
FP insulin (µUI/mL)	10.8 (8.3–15.7)	10.2 (7.7–14.2)	9.0 (6.8–12.0) ***	8.6 (6.7–11.6) **
Increase BL	6.5 ± 13.2	8.6 ± 20.2	5.9 ± 14.4	6.2 ± 14.9
Change BL	34 ± 6	34 ± 5	29 ± 7	26 ± 4
HOMA-IR	2.3 (1.7–3.3)	2.3 (1.7–3.4)	1.9 (1.4–2.6) *	1.9 (1.5–2.8) *
Increase BL	0.7 ± 4.7	0.1 ± 5.9	0.5 ± 3.9 *	−0.1 ± 4.2 *
Change BL (%)	9 ± 7	5 ± 8	7 ± 3	−6 ± 9 *
Adiponectin (µg/mL)	17.0 (12.1–23.0)	16.2 (11.2–22.6)	15.5 (12.3–18.8)	15.3 (11.2–18.9)
Increase BL	−2.9 ± 7.1	−3.8 ± 7.8	−2.5 ± 6.9	−3.0 ± 6.1
Change BL (%)	−10 ± 3	−10 ± 4	−10 ± 3	−10 ± 3
IL-6 (pg/mL)	2.9 (1.9–7.1)	3.0 (1.7–7.5)	1.3 (0–4.8) ***	1.6 (0.1–6.9) **
Increase BL	0.8 ± 5.7	0.3 ± 8.2	−1.1 ± 7.1	−0.8 ± 15.0
Change BL (%)	16 ± 48.4	36.8 ± 19.3	24 ± 21.4	65 ± 21.7
Leptin (ng/mL)	11.8 (7.6–19.5)	7.5 (3.4–15.3)	9.5 (6.3–14.7) **	7.4 (4.2–11.5)
Increase BL	5.0 ± 6.6	0.5 ± 6.6	3.1 ± 5.2 **	0.6 ± 4.8 *
Change BL (%)	9 ± 2	2 ± 1	8 ± 2	4 ± 10
TNF-α (pg/mL)	2.9 (1.9–4.5)	3.3 (2.2–4.4)	2.7 (1.8–3.6)	2.8 (1.8–4.2)
Increase BL	1.9 ± 2.5	1.1 ± 1.2	0.9 ± 1.9	0.8 ± 2.3
Change BL (%)	−8 ± 9	−10 ± 6	25 ± 2	27 ± 4
miR-222-3p	1.04 (−0.5–1.93)	0.85 (−0.11–1.64)	1.29 (0.71–1.87) ***	1.11 (0.62–1.98) ***
miR-103a-3p	−0.12 (−1.48–1.20)	0.35 (−0.92–1.80)	1.57 (0.41–2.47) ***	2.02 (0.52–2.80) ***
miR-132-3p	0.25 (−0.2–0.89)	0.34 (−0.1–0.81)	0.17 (−0.51–0.83)	−0.03 (−0.58–0.86)
miR-29a-3p	0.09 (−0.82–1.43)	0.28 (−0.75–1.41)	0.29 (−1.01–1.93)	0.06 (−1.14–1.77)

Data are mean ± SDM, median IQR or number (%); control group, MedDiet based, with limitation in the consumption of EVOO and nuts; intervention group, MedDiet-based supplemented with EVOO (>40 mL/day) and pistachios (>35 g/day); FP: fasting plasma; OGTT: oral glucose tolerance test; sBP: systolic blood pressure; dBP: diastolic blood pressure; HbA1c-IFCC: haemoglobin glycated, International Federation of Clinical Chemistry; BL: from baseline; TSH: thyroid releasing hormone; FT4: free thyroxin 4; FP insulin: fasting plasma insulin; HOMA-IR: homeostasis model assessment of insulin resistance index; IL-6: interleukin 6; TNF-α: tumour necrosis factor-alpha; miR: microRNA; GW: gestational week; PD: post-delivery; * *p* < 0.05; ** *p* < 0.01 and *** *p* < 0.001 denote differences between both cohort.

**Table 6 nutrients-14-04712-t006:** Body composition and metabolic syndrome components, 2–3 years post-delivery.

	NGT *n* = 236	GDM *n* = 77
BMI (Kg/m^2^)	24.2 ± 4.6	25.0 ± 4.7 *
WC (cm)	81.4 ± 11.1	84.7 ± 12.8 *
Estimated Resting Energy (Kcal/day)	1385 ± 176	1402 ± 225
Total Energy (Kcal/day)	2343 ± 325	2352 ± 276
Fat Mass (kg)	21.3 ± 9.1	22.7 ± 9.0
Lean mass (kg)	41.4 ± 6.0	41.9 ± 6.4
Skeletal Muscle Mass (kg)	18.9 ± 5.4	19.1 ± 3.4
Body Water (L)	31.2 ± 4.9	32.3 ± 5.9
BMI >25 (Kg/m^2^)	100 (42.5)	44 (57.1) *
WC > 89.5 cm	56 (23.7)	32 (41.6) **
Abnormal Glucose Regulation	28 (10.6)	25 (36.4) ***
IFG	21 (8.9)	24 (31.2) ***
IGT	6 (2.5)	10 (12.5) **
Prediabetes (A1c > 5.7%)	11(4.8)	15 (19.7) ***
sBP > 130 mm Hg	3 (1.2)	0
dBP > 85 mm Hg	29 (12.4)	11 (13.8)
TG > 150 mg/dL	12 (5.1)	9 (11.7) *
HDL < 45 mg/dL	25 (10.6)	8 (10.5)
Metabolic Syndrome	26 (11.0)	18 (23.4) *

Data are mean ± SDM or number (%); BMI: body mass index; WC: waist circumference; IFG: impaired fasting glucose; IGT: impaired glucose tolerance; sBP: systolic blood pressure; dBP: diastolic blood pressure; TG: triglyceride; HDL: high-density lipoprotein; GDM: gestational diabetes mellitus; NGT: normal glucose tolerance; * *p* < 0.05, ** *p* < 0.01 and *** *p* < 0.001 denote differences between both cohorts.

**Table 7 nutrients-14-04712-t007:** Statistically significant correlations of miR-222-3p and miR-103a-3p with insulin, HOMA-IR, adipokines and cytokines.

	Crude								Adjusted for ’Pre-Pregnancy BMI (kg/m^2^)							
	miR-222-3p				miR-103a-3p				miR-222-3p				miR103a-3p			
	24–28 GW		2–3 Yrs PD		24–28 GW		2–3 Yrs PD		24–28 GW		2–3 Yrs PD		24–28 GW		2–3 Yrs PD	
	R	*p*-Value	R	*p*-Value	R	*p*-Value	R	*p*-Value	R	*p*-Value	R	*p*-Value	R	*p*-Value	R	*p*-Value
Pre-pregnancy BMI (kg/m^2^)	0.111	0.079	**0.155**	**0.014**	0.081	0.198	**0.13**	**0.039**								
Insulin µUI/mL																
FP 24 GW	**−0.127**	**0.044**	−0.108	0.091	**−0.165**	**0.008**	**−0.148**	**0.018**	**−0.157**	**0.013**	**−0.152**	**0.018**	**−0.189**	**0.003**	**−0.184**	**0.003**
Change at 24 GW	**0.137**	**0.03**	0.087	0.174	**0.151**	**0.016**	**0.162**	**0.01**	**0.151**	**0.017**	0.106	0.1	**0.162**	**0.01**	**0.179**	**0.005**
Change at 2 yrs PD	0.092	0.146	0.026	0.686	0.111	0.078	0.116	0.066	0.118	0.064	0.057	0.376	**0.131**	**0.037**	**0.146**	**0.02**
HOMA-IR																
24 GW	−0.119	0.06	−0.098	0.125	**−0.152**	**0.015**	**−0.133**	**0.035**	**−0.153**	**0.016**	**−0.147**	**0.022**	**−0.179**	**0.004**	**−0.173**	**0.006**
Change at 24 GW	**0.128**	**0.043**	0.078	0.222	**0.149**	**0.018**	**0.155**	**0.014**	**0.144**	**0.023**	0.099	0.122	**0.161**	**0.01**	**0.174**	**0.006**
Change at 2 yrs PD	0.079	0.211	0.025	0.701	0.103	0.1	0.108	0.087	0.105	0.099	0.056	0.385	**0.124**	**0.049**	**0.139**	**0.028**
Leptin (ng/mL)																
12 GW	**0.156**	**0.025**	**0.193**	**0.006**	0.022	0.75	0.093	0.18	0.08	0.25	0.093	0.188	−0.074	0.283	0.001	0.989
24 GW	−0.028	0.693	0.024	0.732	**−0.149**	**0.03**	−0.12	0.083	−0.117	0.093	−0.089	0.209	**−0.247**	**<0.001**	**−0.226**	**0.001**
2 yrs PD	0.029	0.677	0.09	0.201	−0.02	0.775	0.019	0.786	−0.103	0.143	−0.064	0.37	**−0.148**	**0.032**	−0.114	0.099
Change at 24 GW	**0.23**	**<0.001**	**0.239**	**<0.001**	**0.202**	**0.003**	**0.269**	**<0.001**	**0.2**	**0.004**	**0.199**	**0.005**	**0.176**	**0.011**	**0.242**	**<0.001**
Change at 2 yrs PD	**0.154**	**0.026**	**0.146**	**0.038**	0.032	0.646	0.098	0.155	**0.18**	**0.01**	**0.177**	**0.012**	0.049	0.477	0.121	0.081
Adiponectin (µg/mL)																
2 yrs PD	−0.123	0.104	**−0.163**	**0.032**	−0.034	0.649	−0.064	0.394	−0.093	0.221	−0.122	0.112	−0.001	0.989	−0.022	0.772
TNF-α (pg/mL)																
12GW	−0.075	0.281	0.087	0.217	**−0.16**	**0.02**	−0.055	0.424	−0.09	0.198	0.071	0.316	**−0.172**	**0.012**	−0.067	0.333
24GW	**−0.14**	**0.044**	0.021	0.762	**−0.204**	**0.003**	−0.102	0.14	**−0.153**	**0.028**	0.005	0.949	**−0.215**	**0.002**	−0.111	0.108
IL-6 (pg/mL)																
12 GW	**−0.152**	**0.028**	−0.008	0.911	**−0.243**	**<0.001**	**−0.147**	**0.033**	**−0.151**	**0.031**	−0.006	0.935	**−0.242**	**<0.001**	**−0.145**	**0.037**
24 GW	**−0.151**	**0.029**	0	0.995	**−0.307**	**<0.001**	**−0.156**	**0.023**	**−0.165**	**0.018**	−0.016	0.826	**−0.318**	**<0.001**	**−0.168**	**0.015**
2 yrs PD	−0.092	0.184	−0.003	0.971	**−0.238**	**<0.001**	**−0.14**	**0.042**	−0.106	0.128	−0.019	0.784	**−0.25**	**<0.001**	**−0.154**	**0.026**

GW = Gestational week; yrs PD = years post-delivery; IL-6 = interleukin 6; BMI = body mass index; TNF-α = Tumour necrosis factor-alpha; HOMA-IR = Homeostasis model assessment of insulin resistance index; Spearman’s rho; change = level change (%) from baseline. **Highlighted in bold *p* values < 0.001**.

## Data Availability

All relevant data are within the paper and its Appendix A.

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
