# Peer review of "The Relationship between Serum Adipokines, miR-222-3p, miR-103a-3p and Glucose Regulation in Pregnancy and Two to Three Years Post-Delivery in Women with Gestational Diabetes Mellitus Adhering to Mediterranean Diet Recommendations"

_nutrients, 2022, doi:10.3390/nu14224712_

Round 1
Reviewer 1 Report (New Reviewer)
I found the manuscript interesting and well-targeted to the journal. However, I believe that some improvements are required. Please consider the following points:
A) Although you declared in the methods the assessment of distribution normality of continuous variables, I am not sure that all the variables described by mean and standard deviation are normally distributed (e.g., 2.18 ± 3.38 - Table 1). Hence I believe that a statistical revision is required.
B) English language needs to be improved. For example, BW is usually used in fetal medicine as birth weight, not body weight. Please amend. In addition, other typos are present such as "adiponectin and insulin resistance that that persisted" (page 3). Please carefully revise the English language.
Author Response
Comments and Suggestions for Authors
I found the manuscript interesting and well-targeted to the journal. However, I believe that some improvements are required. Please consider the following points:
A) Although you declared in the methods the assessment of distribution normality of continuous variables, I am not sure that all the variables described by mean and standard deviation are normally distributed (e.g., 2.18 ± 3.38 - Table 1). Hence I believe that a statistical revision is required.
- We agree with your comments. We had only applied the normality test to the study variables. We modified the values ​​of body weight in median and interquartile range and the comparison with the Mann-Whitney U in Table 1 and 2. The statistical data, as you can see, are not modified.
- B) English language needs to be improved. For example, BW is usually used in fetal medicine as birth weight, not body weight. Please amend. In addition, other typos are present such as "adiponectin and insulin resistance that that persisted" (page 3). Please carefully revise the English language.
- Following your comment we have replaced BW with Body weight to avoid confusion in tables 4 and 5 and that has been deleted.

Reviewer 2 Report (New Reviewer)
The article investigates a possible correlation between a specific dietary regime in early pregnancy, the expression of leptin, adiponectin, and pro-inflammatory cytokines, and the expression different signalling molecules and miRNAs in GDM.
Dealing with possible predictive factors for the development of GDM the article is of high relevance with respect to diabetes research and prevention, and should therefore be published.
I like that you included the calculation of sample size in the M&M.
Wording / presentation
Introduction
- „GDM adiposity“? This is not clear to me. Did you mean GDM and adiposity? GDM in adipose women? Adipose kids after GDM?
M&M
- “and parity”: What exactly is the parity of women? Assuming you meant that both groups were comparable concerning mean age/weight of the participants please rephrase, it really reads like “according to their age, their weight, and their parity”
- TSH: Please also include the full name with the first mention in the text, not only below a table. Same goes for FT4
- If you include the city of a company only once (sensible), please do so at the 1st mention
- “X2 or ANOVA”: So which did you use for what?
Results
- Either reword or at least put a semicolon after the 1st half of the 1st sentence, I had to read that one 3 times to understand it
- I personally would find it helpful if the significant differences were indicated in some way in all tables, (you’ve done so in table for, but not table 2 for example)
- Also please leave a space between table description and normal text
- Also, just a matter of looks: could you use the same font in all your tables please?
General questions
If both groups adhered to a generally healthy nutritional regime, but the control group had to restrict their general fat intake, could the results not also be due to a lack of fats (and therefore, amongst other things, a reduced intake of fat-soluble vitamins), instead of the increased intake of olive oil and pistachios?
Comparing the table and the description “demographic data” leads me to conclude that there was not a single woman that did smoke in the past but stopped when she found out about the pregnancy. is this true? Otherwise, this information is either missing in the table or misleadingly called “current”
Author Response
Comments and Suggestions for Authors
The article investigates a possible correlation between a specific dietary regime in early pregnancy, the expression of leptin, adiponectin, and pro-inflammatory cytokines, and the expression different signalling molecules and miRNAs in GDM.
Dealing with possible predictive factors for the development of GDM the article is of high relevance with respect to diabetes research and prevention, and should therefore be published.
I like that you included the calculation of sample size in the M&M.
- The estimation of the sample size is referred to on page 3, section 2.2.
Wording / presentation
Introduction
- „GDM adiposity“? This is not clear to me. Did you mean GDM and adiposity? GDM in adipose women? Adipose kids after GDM?
- GDM and adiposity
M&M
- “and parity”: What exactly is the parity of women? Assuming you meant that both groups were comparable concerning mean age/weight of the participants please rephrase, it really reads like “according to their age, their weight, and their parity”
We have rephrased: … according to their ethnic origin, their age, their weight, and their parity. -
TSH: Please also include the full name with the first mention in the text, not only below a table. Same goes for FT4
Thyroid Stimulating Hormone (TSH) and free thyroxine (FT) are inserted.
- If you include the city of a company only once (sensible), please do so at the 1st mention
We have done it as requested.
- “X2 or ANOVA”: So which did you use for what?
ANOVA has been deleted. It has been clarified: characteristics for categorical variables were evaluated by the χ2 test . For continuous variables, measures were compared with Student’s t test or the Mann–Whitney U test if distribution of quantitative variables were or not normal, respectively.
Results
- Either reword or at least put a semicolon after the 1st half of the 1st sentence, I had to read that one 3 times to understand it
To improve understanding, the percentage of both GDM and NGT women in the intervention group has been suppressed.
A total of 77 (24.6%) women developed GDM, 37 (48.1%) from Intervention Group while 236 (75.4%) were NGT, 135 (57.2%) from the Intervention Group.
- I personally would find it helpful if the significant differences were indicated in some way in all tables, (you’ve done so in table for, but not table 2 for example)
As the tables have 4 columns, and comparisons are made between 2 groups (columns), the statistical differences are referred to with symbols to prevent the table from being too wide.
- Also please leave a space between table description and normal text
A space has been inserted
- Also, just a matter of looks: could you use the same font in all your tables please?
The font in all tables has been unified.
General questions
If both groups adhered to a generally healthy nutritional regime, but the control group had to restrict their general fat intake, could the results not also be due to a lack of fats (and therefore, amongst other things, a reduced intake of fat-soluble vitamins), instead of the increased intake of olive oil and pistachios?
Indeed, there may be confounding factors that have not been controlled in this study. We have included this phrase in limitations: On the other hand, differences in the intake of fat-soluble vitamins, which have not been evaluated in the study, could also affect the results.
Comparing the table and the description “demographic data” leads me to conclude that there was not a single woman that did smoke in the past but stopped when she found out about the pregnancy. is this true? Otherwise, this information is either missing in the table or misleadingly called “current”
As in section 2.4.1, the habit of smoking is recorded in 3 levels: those who state that they have never been smokers (NEVER) - those who state that they have been smokers until they knew they were pregnant - and those who state that they continue to smoke (CURRENT). In the table only the first and last are shown.

Reviewer 3 Report (Previous Reviewer 1)
This manuscript aims to explore the relationship of leptin, adiponectin, IL-6, TNF-α, insulin, and HOMA-IR and circulating miRNAs (miR-29a-3p, miR-103a-3p, miR-132-3p, miR-222-3p) with the appearance of GDM and with Med-Diet-based nutritional intervention, at 24-28 GW, and in glucose regulation 2-3 years PD. The study is interesting.
1. What are the hypotheses for studying the correlation between circulating levels of miRNAs and adipokines, cytokines, insulin sensitivity? This description is inconsistent before and after in the manuscript. It was concluded in the abstract as "These data support the association of leptin, adiponectin and insulin / HOMA-IR with GDM, as well as the association of insulin / HOMA-IR and IL-6 and mir-222-3p and mir-103a-3p expression with a meddiet based nutritional intervention. " However, table 4 shows that only 2-3 years postpartum the GDM group had higher mir-222-3p levels than the NGT group. Does this suggest a nonsignificant association between mir-103a-3p and GDM? Moreover, whether mir-222-3p overexpression promotes or inhibits GDM progression? In addition, table 5 shows that mir-103a-3p with mir-222-3p expression was higher in the Mediterranean diet intervention group than in the control group, and table 7 shows that mir-103a-3p and mir-222-3p levels were negatively correlated with insulin, HOMA-IR, adiponectin, and inflammatory cytokine levels. It appears then that there may not be a causal association between miRNA expression levels and changes in adipokines, inflammatory cytokines, insulin sensitivity in GDM, but rather a consequence of Mediterranean dietary intervention with this common factor. The author's elaboration of the logical hypothesis for this association is not clear enough, and no clear conclusions can be drawn, leading to a confusing logical relationship for this important conclusion. It is recommended to tease out clear research hypotheses and draw scientific conclusions around the findings elucidating their logical relationships.
2. The methods section should be described clearly. Does the duration of interventions in studies of the Mediterranean diet to prevent GDM last until the diagnosis of GDM at 24-28 weeks' gestation, or does it continue until the delivery phase? If continued until the delivery stage, Mediterranean dietary interventions are themselves important and impactful confounders. This affects an accurate analysis of the conclusions of this manuscript.
3. The table for this manuscript involves multiple ways of grouping. To enable a clear and accurate understanding of the results and conclusions of this manuscript, it is recommended that each table should be supplemented with corresponding grouping or timing information. For example, table 1 should add gestational weeks 8-12 to the title, along with a description of the control and intervention group specific information in notes.
4. The results of the correlation analysis in Table 7 of the first submission are positively correlated. Why was this negative in the second submission?
5. The conclusion of the manuscript should be a summary, not an expanded discussion. For example, "pistachios and EVOO are rich in unsatured fatty acids, magnesium, and other biochemical complications with potential beneficial effects on insulin sensitivity, fasting glucose levels, and inflammation. We hypothesize that they have antibiotic capacity, given their high levels of lutein, β- carotene, and γ- tocopherol, may enhance miR-103a-3p and miR-222-3p overexpression and improve the inflammatory cytokine profiles linked to GDM development ” This part should be placed under discussion accordingly.
Author Response
Comments and Suggestions for Authors
This manuscript aims to explore the relationship of leptin, adiponectin, IL-6, TNF-α, insulin, and HOMA-IR and circulating miRNAs (miR-29a-3p, miR-103a-3p, miR-132-3p, miR-222-3p) with the appearance of GDM and with Med-Diet-based nutritional intervention, at 24-28 GW, and in glucose regulation 2-3 years PD. The study is interesting.
- What are the hypotheses for studying the correlation between circulating levels of miRNAs and adipokines, cytokines, insulin sensitivity? This description is inconsistent before and after in the manuscript. It was concluded in the abstract as "These data support the association of leptin, adiponectin and insulin / HOMA-IR with GDM, as well as the association of insulin / HOMA-IR and IL-6 and mir-222-3p and mir-103a-3p expression with a meddiet based nutritional intervention. " However, table 4 shows that only 2-3 years postpartum the GDM group had higher mir-222-3p levels than the NGT group. Does this suggest a nonsignificant association between mir-103a-3p and GDM? Moreover, whether mir-222-3p overexpression promotes or inhibits GDM progression? In addition, table 5 shows that mir-103a-3p with mir-222-3p expression was higher in the Mediterranean diet intervention group than in the control group, and table 7 shows that mir-103a-3p and mir-222-3p levels were negatively correlated with insulin, HOMA-IR, adiponectin, and inflammatory cytokine levels. It appears then that there may not be a causal association between miRNA expression levels and changes in adipokines, inflammatory cytokines, insulin sensitivity in GDM, but rather a consequence of Mediterranean dietary intervention with this common factor. The author's elaboration of the logical hypothesis for this association is not clear enough, and no clear conclusions can be drawn, leading to a confusing logical relationship for this important conclusion. It is recommended to tease out clear research hypotheses and draw scientific conclusions around the findings elucidating their logical relationships.
The aim of this study is to analyze the mechanisms involved in gestational diabetes development and in the protective effects of a Mediterranean diet supplemented with extra virgin olive oil and pistachios, as compared to standard treatment in pregnant women. For this reason we have explored different pathways that might be involved: an inflammatory pathway: cytokines (TNF-α and IL-6), a lipostatic pathway: adipokines (leptin and adiponectin) and an epigenetic pathway: miRNAs expression. We think it is interesting to evaluate whether these pathways have an independent effect or there is a correlation and an additive effect.
We agree with your comment that we have not found and association between miR-222 and miR-103 and GDM diagnosis at 24-28 GW. It seems that GDM risk is more related to a deleterous lipostatic profile (higher leptin/lower adiponectin) and to a well known insulin resistance. Nevertheless, an intervention with a Mediterranean Diet is associated with and increased expression of these miRNAs and an improvement in the inflammatory (decreased IL-6) and insulin sensitivity profiles. The finding of an association between a nutritional intervention during pregnancy and these epigenetic changes is really interesting, since we have previously shown in this cohort, that an intervention based on the MedDiet during pregnancy is associated with a reduction in offspring’s hospital admissions in the first 2 years of age, due to bronchiolitis/asthma or other diseases that required either antibiotic or corticosteroid treatment (J. Clin. Med. 2020, 9, 1454; doi:10.3390/jcm9051454).
We do not estate that there is a causal association between miRNA expression levels and changes in adipokines, cytokines and insulin sensitivity in GDM. On the contrary, we agree with you that these changes are likely to be related to the nutritional intervention with MedDiet.
For these reasons, we truly believe that our data support our conclusion of an association between GDM and deleterous lipostatic + insulin sensitivity profiles on one hand, and an association between an intervention with MedDiet and improvements in insulin sensitivity and inflammatory profile, and epigenetic changes, on the other.
- The methods section should be described clearly. Does the duration of interventions in studies of the Mediterranean diet to prevent GDM last until the diagnosis of GDM at 24-28 weeks' gestation, or does it continue until the delivery phase? If continued until the delivery stage, Mediterranean dietary interventions are themselves important and impactful confounders. This affects an accurate analysis of the conclusions of this manuscript.
According to your comments we have described in a more complete way the nutritional intervention:
In summary, all the women received instructions and recommendations on the Mediterranean diet from the beginning of pregnancy, prior to GW 12. The women included in the Intervention Group (IG: MedDiet-based supplemented with EVOO and pistachios) were recommended to increase the consumption of EVOO (>40 ml/day) and pistachios (>35 g daily), for which they were supplied free of charge with a 5 l carafe of EVOO and 1 Kg of pistachios every 2 months until delivery, to guarantee reaching the minimum consumption. The women included in the control group (CG: MedDiet based, with limitation in the consumption of EVOO and nuts) received the same nutritional recommendations except to limit the consumption of EVOO (<40 ml/day) and nuts of any sort (<25 g/day). These recommendations were maintained until delivery. After birth, all women received the same nutritional recommendations based on the MedDiet, but without restricting the consumption of EVOO or nuts, nor were EVOO or nuts provided free of charge.
It is possible that the few differences in the nutritional pattern between the two groups might affect the results, and therefore it is referred to as one of the limitations of the study. However, for this reason, the results are always analyzed by intention to treat, avoiding possible bias.
- The table for this manuscript involves multiple ways of grouping. To enable a clear and accurate understanding of the results and conclusions of this manuscript, it is recommended that each table should be supplemented with corresponding grouping or timing information. For example, table 1 should add gestational weeks 8-12 to the title, along with a description of the control and intervention group specific information in notes.
Following your comments we have included in tables 1 and 2 the baseline gestation week and in the footnote the identification of cG (CG: MedDiet based, with limitation in the consumption of EVOO and nuts) and iG (IG: MedDiet-based supplemented with EVOO and pistachios).
- The results of the correlation analysis in Table 7 of the first submission are positively correlated. Why was this negative in the second submission?
The substantial difference between the initial version and the 1st revision form, and the current version is that the correlation was initially established considering a normal distribution of the miR-222 and miR-103 expression with Pearson correlation coefficient, and later the R Spearman's correlation test was used, when considering a non-normal distribution.
- The conclusion of the manuscript should be a summary, not an expanded discussion. For example, "pistachios and EVOO are rich in unsatured fatty acids, magnesium, and other biochemical complications with potential beneficial effects on insulin sensitivity, fasting glucose levels, and inflammation. We hypothesize that they have antibiotic capacity, given their high levels of lutein, β- carotene, and γ- tocopherol, may enhance miR-103a-3p and miR-222-3p overexpression and improve the inflammatory cytokine profiles linked to GDM development ” This part should be placed under discussion accordingly.
Following your recommendation, we have changed the summary paragraph at the end of the discussion and have included this short sentence in the conclusion:
In conclusion, the nutritional intervention based on the principles of the Mediterranean Diet is associated with an overexpression of miR-222 and miR-103 that correlates with an improvement in insulin sensitivity and in the inflammatory profile of cytokines

Round 2
Reviewer 1 Report (New Reviewer)
The authors adequately addressed the previous comments, and I found the manuscript improved and suitable to be published.
This manuscript is a resubmission of an earlier submission. The following is a list of the peer review reports and author responses from that submission.
Round 1
Reviewer 1 Report
This study was designed to be based on a RCT of a Mediterranean dietary intervention for the prevention of gestational diabetes with follow-up until 2-3 years post-delivery (PD).. This manuscript analyzes changes in adipokines and the serum expression of miRNAs during the first trimester, at GDM screening, and at 2-3 years PD. And to analyze the associations of these changes with the risk of developing GDM, the risk of developing postpartum abnormal glucose regulation (AGR),, and the effects of Mediterranean dietary interventions. This study is interesting.
However, the following aspects exist for this manuscript for clarity.
1. The purpose of this study is scattered and unclear. The authors planned to explore more diverse issues, including the predictive effect of adipokines and the serum expression of miRNAs on the risk of developing GDM, the predictive effect of adipokines and the serum expression of miRNAs on the risk of developing AGR, and so on. However, the phase I study was an RCT study of the Mediterranean diet. Because there is an effect of the Mediterranean diet, an important intervention, this study type was not appropriate to perform the predictive effect of adipokines and the serum expression of miRNAs on the risk of developing GDM. By the same token, adipokines and the serum expression of miRNAs in phase I are also not suitable for predictive effects on the risk of the occurrence of AGR.
2. The authors analyzed the association between adherence to the Mediterranean diet, the risk of AGR occurring 2-3 years PD, adipokines, and the serum expression of miRNAs. However, the risk of developing GDM differed between the Mediterranean diet intervention group and the control group, and GDM had effects on levels of adipokines. Therefore, this factor may interfere with the results of the association analysis between Mediterranean diet adherence and adipokines.
3. Because the data in this manuscript were obtained from a phase I RCT study and a follow-up observational study in phase II, the statistical analysis protocol was confusing. There were several methods of subgroup analysis (intervention group vs control group, GDM group vs NGT group, and AGR group vs NGT group). Such a statistical analysis scheme affects the strength and logic of the argument for the purpose of the study.
4. The results of Mediterranean diet adherence in the control and intervention groups in Table 3 are doubtful. In the control group, adherence to the Mediterranean diet was 4.5 ± 1.6 versus 5.2 ± 1.9 for NGT versus GDM at 24-28 gestational weeks, respectively. The reviewer re calculated that indeed the difference was statistically significant. The difference was also statistically significant at 2-3 years PD, which was lower in the NGT group than in the GDM group. Is this due to the emphasis on a Mediterranean diet among pregnant women who diagnose GDM? Adherence to the Mediterranean diet was higher in the GDM group. What is the interactive effect of two factors, Mediterranean diet adherence and GDM, on outcomes? Does it affect the interpretation of conclusions?
5. The author should put forward a clear scientific hypothesis in Introduction about the relationship and internal mechanism between leptin, adiponectin, inter-leukin-6 (IL-6), tumour necrosis factor-alpha (TNF-α), insulin, and HOMA-IR, and circulating miRNAs (miR-29a-3p, miR-103-3p, miR-132-3p, miR-222-3p). In addition, the study results mainly list the results of these indicators, and it is suggested to carry out further statistical analysis and discussion on the intrinsic mechanism of these indicators.
Author Response
Rev 1
Thank you very much for your constructive comments. We will answer each of your questions:
- The purpose of this study is scattered and unclear. The authors planned to explore more diverse issues, including the predictive effect of adipokines and the serum expression of miRNAs on the risk of developing GDM, the predictive effect of adipokines and the serum expression of miRNAs on the risk of developing AGR, and so on. However, the phase I study was an RCT study of the Mediterranean diet. Because there is an effect of the Mediterranean diet, an important intervention, this study type was not appropriate to perform the predictive effect of adipokines and the serum expression of miRNAs on the risk of developing GDM. By the same token, adipokines and the serum expression of miRNAs in phase I are also not suitable for predictive effects on the risk of the occurrence of AGR.
Response: We agree that the San Carlos study was designed to evaluate the effect of a nutritional intervention based on the Mediterranean diet supplemented with EVOO and pistachios, and applied early from the start of pregnancy (before GW 12) could reduce the appearance of GDM. St GDM prevention study demonstrated a 30% reduction in the rate of GDM. After delivery, the women included in the study received identical nutritional recommendations without providing free EVOO or pistachios, and between 2 and 3 years postpartum they were evaluated for postnatal abnormal glucose regulation.
.There is increasing evidence that miRNAs participate in the pathogenesis of diabetes and different miRNAs are expressed differently in pancreatic beta cells, liver, muscle and adipose tissue and the placenta.
.Variations in epigenetic programming can be caused by environmental factors such as diet, through the regulation of gene expression by miRNAs
.These miRNAs (circulating miRNAs (miR-29a-3p, miR-103-3p, miR-132-3p, miR-222-3p) affect different proinflammatory and adipogenic pathways and insulin signaling through different adipokines leptin, adiponectin, inter-leukin-6 (IL-6), tumor necrosis factor-alpha (TNF-α), insulin, and HOMA- IR.
. The altered expression of miRNAs prior to glycemic alteration both during pregnancy and postnatal life allows us to hypothesize about their participation in the development of GDM and postnatal AGR as well as the influence of nutritional intervention early during pregnancy
For this reason, we believe that the St Carlos GDM prevention study represents an opportunity to evaluate these aspects. Following their indications and comments, we reformulate the hypothesis and objectives in the introduction by inserting this paragraph.
The hypothesis of the current study is that circulating levels of aforementioned miRNA could be biomarkers for the appearance of GDM and postnatal AGR, acting through the modification of adipokines and insulin sensitivity. Nutritional intervention based on the Mediterranean diet can prevent changes in the circulating levels of these miRNAs and adipokines and thus reduce the rate of appearance of GDM and postnatal AGR. (CITA) This study therefore assesses the adipokines (leptin and adiponectin), and inflammatory cytokines (IL-6 and TNF-α), as well as the serum expression of miRNAs (miR-29a-3p, miR-103-3p, miR-132-3p and miR-222-3p), at the beginning of gestation (visit 1, baseline 8-12 GW), at the time of GDM screening (visit 2: 24-28 GW) and at the moment of the 2-3 year follow-up when postpartum abnormal glucose regulation (AGR) may be diagnosed (visit 3).
And we reformulate the objectives
The primary objective is to assess whether adipokines/circulating miRNAs levels differed between women from the Mediterranean diet intervention group and those from the control group. The secondary objective is to detect the differences between women who develop GDM and/or postnatal AGR and those who developed neither condition, in an attempt to identify possible early biomarkers of glucose dysregulation.
The authors analyzed the association between adherence to the Mediterranean diet, the risk of AGR occurring 2-3 years PD, adipokines, and the serum expression of miRNAs. However, the risk of developing GDM differed between the Mediterranean diet intervention group and the control group, and GDM had effects on levels of adipokines. Therefore, this factor may interfere with the results of the association analysis between Mediterranean diet adherence and adipokines.
Response: Analyzes between groups were considered by intention to treat and not by protocol (adherence). In this way it is recognized that confusion bias is avoided. Thus, the women from the intervention group were compared with those from the control group (table 5) and not only by the presence of disease or not (GDM vs NGT) in table 4
All primary analyses were performed on an intention-to-treat basis, IG vs CG no per protocol
- Because the data in this manuscript were obtained from a phase I RCT study and a follow-up observational study in phase II, the statistical analysis protocol was confusing. There were several methods of subgroup analysis (intervention group vs control group, GDM group vs NGT group, and AGR group vs NGT group). Such a statistical analysis scheme affects the strength and logic of the argument for the purpose of the study.
Response: The sample size was calculated to observe differences between the control group compared to the intervention group. And the primary analysis was carried out by intention to treat and not by complying with the protocol or adherence. However, it was also analyzed by the presence of disease in relation to its absence (GDM vs NGT and AGR vs NGT). Indeed, there may be interference and confounding factors, but this does not detract from the value of the comparative study
- The results of Mediterranean diet adherence in the control and intervention groups in Table 3 are doubtful. In the control group, adherence to the Mediterranean diet was 4.5 ± 1.6 versus 5.2 ± 1.9 for NGT versus GDM at 24-28 gestational weeks, respectively. The reviewer re calculated that indeed the difference was statistically significant. The difference was also statistically significant at 2-3 years PD, which was lower in the NGT group than in the GDM group. Is this due to the emphasis on a Mediterranean diet among pregnant women who diagnose GDM? Adherence to the Mediterranean diet was higher in the GDM group. What is the interactive effect of two factors, Mediterranean diet adherence and GDM, on outcomes? Does it affect the interpretation of conclusions?
Response:
Indeed there is an error in the value of MedS in 24 GW in the value of the table. The analysis is as follows:
Estadísticos de grupo |
|||||
|
|||||
|
|||||
|
DIABETESGESTACIONAL |
N |
Media |
Desviación típ. |
Error típ. de la media |
COMPUTE SG24_MEDDIET_SCORE |
SI |
40 |
4,8750 |
2,01517 |
,31863 |
NO |
101 |
4,4646 |
1,58638 |
,15716 |
Prueba de muestras independientes |
||||||||||
|
Prueba de Levene para la igualdad de varianzas |
Prueba T para la igualdad de medias |
||||||||
F |
Sig. |
T |
gl |
Sig. (bilateral) |
Diferencia de medias |
Error típ. de la diferencia |
95% Intervalo de confianza para la diferencia |
|||
Inferior |
Superior |
|||||||||
COMPUTE SG24_MEDDIET_SCORE |
Se han asumido varianzas iguales |
1,150 |
,485 |
2,426 |
135 |
,077 |
,17220 |
,11826 |
,14277 |
1,40162 |
No se han asumido varianzas iguales |
|
|
2,257 |
58,666 |
,088 |
,17220 |
,14216 |
,08745 |
1,45695 |
The differences are not significant
- The author should put forward a clear scientific hypothesis in Introduction about the relationship and internal mechanism between leptin, adiponectin, inter-leukin-6 (IL-6), tumour necrosis factor-alpha (TNF-α), insulin, and HOMA-IR, and circulating miRNAs (miR-29a-3p, miR-103-3p, miR-132-3p, miR-222-3p). In addition, the study results mainly list the results of these indicators, and it is suggested to carry out further statistical analysis and discussion on the intrinsic mechanism of these indicators.
Response: The hypothesis and the objectives have been reformulated (Q1)
In conclusion insert: Pistachios and EVOO are rich in unsaturated fatty acids, magnesium, and other phytochemical constituents with potential beneficial effects on insulin sensitivity, fasting glucose levels and inflammation. We hypothesize that their antioxidant capacity given their high levels of lutein,β-carotene, and γ-tocopherol may enhance mir103-p and mir222-3p overexpression and improves the inflammatory cytokine profiles linked to GDM development.
In addition, as suggested, a linguistic revision has been performed by a Native American doctor

Reviewer 2 Report
This is an interesting manuscript regarding how MedDiet may influence the GDM status. The aim of the article is to investigate the relationship between serum inflammatory markers and circulating miRNAs (miR-29a-3p, miR-103-3p, miR-132-3p, miR-222-3p) in the appearance of GDM, at 24-28 gestational weeks (GW), and in glucose regulation 2-3 years post-delivery (PD).
1. Please briefing clarify the intervention diet in the abstract. Do the pregnant women receive general dietary guidance or an individual dietary planner? And for how long?
2. Please change the word “involvement” to “association of leptin and TNF-α and miR-222-3p with GDM” in the last line in the abstract.
3. In the conclusion section in the abstract: Do the effects of MedDiet persist over time or do the pregnant women still under this diet? Please clarify.
4. Introduction section, second paragraph: 1. You mean to find “inflammatory" markers of with potential diagnostic and therapeutic utility because we already have serum markers like serum glucose and HOMA. 2. Which components of the MedDiet are related to modulating inflammation?
5. The Nutrition score was evaluated by which tool? Does it have a cutoff point to better understand changes during the intervention?
6. The adherence to physical activity did not present a significant difference among groups in Table 3. How this could be related to the reduction in serum inflammatory markers since physical activity is associated with lower insulin resistance and proinflammatory markers, especially IL-6, at either 24-28 GW or 2-3 years?
7. In Table 8, I suggest that the correlation analysis should be adjusted by HOMA and weight since the miRNA evaluated affects insulin and glucose metabolism.
8. To improve the discussion section, please add information regarding which MedDiet nutritional components may act as regulating miRNA or inflammatory markers. Please add possible action mechanisms.
Author Response
Rev 2
Thank you very much for your constructive comments.
- Pleasebriefing clarify the intervention diet in the abstract. Do the pregnant women receive general dietary guidance or an individual dietary planner? And for how long?
Response: The intervention is entered in the abstract as follows: ... Intervention Group, (MedDiet supplemented with extra virgin olive oil (EVOO) and pistachios during pregnancy) Control Group (MedDiet restrict the consumption of dietary fat including EVOO and nuts) during pregnancy
- 2. Please change the word “involvement” to “association of leptin and TNF-α and miR-222-3p with GDM” in the last line in the abstract.
Response: the word “involvement” has been changed to “association”
- In the conclusion section in the abstract: Do the effects of MedDiet persist over time or do the pregnant women still under this diet?Please clarify
Response: The main difference between the IG and CG is during pregnancy, prior to 12 GW until delivery. Postpartum, women in both groups received the same MedDiet nutritional intervention without limiting “healthy” fat intake. We think it is sufficiently clarified when introducing the type of intervention (Intervention Group, (MedDiet supplemented with extra virgin olive oil (EVOO) and pistachios) Control Group (MedDiet restrict the consumption of dietary fat including EVOO and nuts) during pregnancy)
- Introduction section, second paragraph: 1. You mean to find “inflammatory" markers of with potential diagnostic and therapeutic utility because we already have serum markers like serum glucose and HOMA. 2. Which components of the MedDiet are related to modulating inflammation?
Response (MedDiet) supplemented with extra virgin olive oil (EVOO) and pistachios was associated with a 30% reduction in the incidence of GDM. Experimental studies have shown that components of the MedDiet as lutein,β-carotene, γ-tocopherol hidroxithyrosol, may positively modulate the insulin signalling pathway reducing inflammatory cytokines and adipokines and modifying some microRNAs (miRNA) profiles . We believe that references can be enough
- The Nutrition score was evaluated by which tool? Does it have a cutoff point to better understand changes during the intervention?
Response: Dietary intake and physical activity were evaluated, by applying a semi-quantitative frequency questionnaire, based on the Diabetes Nutrition and Complications Trial (DNCT) study. The Nutrition score is between −12 and 12, and the objective is >5 and the physical activity score is between −3 and 3, and the objective is >0 (The Diabetes and Nutrition Study Group of the Spanish Diabetes Association. Diabetes Nutrition and Complications Trial: Trends in nutritional pattern between 1993 and 2000 and targets of diabetes treatment in a sample of Spanish people with diabetes. Diabetes Care 2004; 27:984-7)
We insert after: …… with the Diabetes Nutrition and Complication Trial (DNCT) questionnaire to obtain the nutrition score (NS) and physical activity score. The Nutrition score is between −12 and 12, and the objective is >5 and the physical activity score is between −3 and 3, and the objective is >0
- The adherence to physical activity did not present a significant difference among groups in Table 3.How this could be related to the reduction in serum inflammatory markers since physical activity is associated with lower insulin resistance and proinflammatory markers, especially IL-6, at either 24-28 GW or 2-3 years?
Response: Mainly, the changes that arise in the circulating levels of miRNA and cytokines are associated with changes in eating patterns and not physical activity score. Pistachio consumption has antioxidant capacity, higher than other nuts, given their high levels of lutein,β-carotene, and γ-tocopherol and it improves the inflammatory cytokine profiles linked to GDM development
- In Table 8, I suggest that the correlation analysis should be adjusted by HOMA and weight since the miRNA evaluated affects insulin and glucose metabolism.
Response: Thank you for your comment. Although we had done it, considering the widest range in the HOMA-IR values, we have only found differences in the thousandths of r without affecting p. We consider not only BW, but also BW change and BMI.
- To improve the discussion section, please add information regarding which MedDiet nutritional components may act as regulating miRNA or inflammatory markers. Please add possible action mechanisms.
Response: Pistachios are rich in unsaturated fatty acids, fiber, magnesium, and other phytochemical constituents with potential beneficial effects on insulin sensitivity, fasting glucose levels and inflammation. Their antioxidant capacity is higher than other nuts, given their high levels of lutein,β-carotene, and γ-tocopherol. Pistachio consumption improves the inflammatory cytokine profiles linked to GDM development.Increased EVOO and pistachio consumption was clearly beneficial. EVOO is a rich source of monounsaturated fatty acids, and has been found to lower postprandial glucose levels as well as to improve the inflammatory profile
In conclusion insert: Pistachios and EVOO are rich in unsaturated fatty acids, magnesium, and other phytochemical constituents with potential beneficial effects on insulin sensitivity, fasting glucose levels and inflammation. We hypothesize that their antioxidant capacity given their high levels of lutein,β-carotene, and γ-tocopherol may enhance mir103-p and mir222-3p overexpression and improves the inflammatory cytokine profiles linked to GDM development.
In addition, as suggested, a linguistic revision has been performed by a Native American doctor

Round 2
Reviewer 1 Report
The authors responded to the review questions accordingly, rewrote the research hypothesis and purpose, and revised the questioned data. The following questions still exist in the revised manuscript.
1. The authors put forward the research hypothesis that "Nutritional intervention based on the Mediterranean diet can prevent changes in the circulating levels of these miRNAs and adipokines and thus reduce the rate of appearance of GDM and postnatal AGR.". The corresponding primary study objective was “to assess whether adipokines/circulating miRNAs levels differed between women from the Mediterranean diet intervention group and those from the control group." However, there was no statistical difference in the incidence of GDM between the control group and the Mediterranean diet intervention group in Table 1. In Table 3, there was no statistically significant difference in Mediterranean diet adherence between participants with and without GDM in the control group and the Mediterranean diet intervention group (both at baseline and 24-28GW). In Table 5, there was no statistically significant difference in fasting blood glucose at 24-28 GW between the control group and the Mediterranean diet intervention group (statistical significance noted by the authors, not statistically significant after reviewer recalculation). Taken together, the results of this manuscript do not support a Mediterranean dietary intervention to prevent the risk of GDM. However, this key is the starting point for this study.
2. The authors also put forward the research hypothesis that "circulating levels of aforementioned miRNA could be biomarkers for the appearance of GDM and postnatal AGR, acting through the modification of adipokines and insulin sensitivity". The stated purpose of the study was "The secondary objective is to detect the differences between women who develop GDM and/or postnatal AGR and those who developed neither condition, in an attempt to identify possible early biomarkers of glucose dysregulation". In Table 4 , the miR-222-3p between the GDM group and the control group at 24-28GW was actually not significantly different (statistical significance noted by the authors, not statistically significant after reviewer recalculation). And there was no significant difference in miR-103-3p, miR-132, miR-29a-3p between the two groups. No data were seen for the statistical description of these indicators between the AGR group and the NGT group. In Table 8, the adjusted correlation analysis results show that there is no statistical correlation between the levels of miR-222-3p and miR-103-3p at 24-28GW and 2-3 years of PD and the 2h GTT level of 2-3 years of PD. These results do not support the conclusions (association of miR222-3p with GDM) presented by the authors.
3. The statistical methodology of this manuscript needs to be re-examined. For example, IL-6, 8-12GW weight gain, HOMA-IR, fasting insulin, miR-222-3p, miR-103-3p, miR-132, and miR-29a-3p did not seem to conform to a normal distribution. A test for normality should be performed and then the correct statistical method for comparison between groups should be reselected. In addition, there are deviations in the statistical analysis results in many places. Corrections should be recalculated to draw correct conclusions.
Author Response
Thank you very much for your interest in our article. We are going to answer the questions that you consider remain unanswered.
- The authors put forward the research hypothesis that "Nutritional intervention based on the Mediterranean diet can prevent changes in the circulating levels of these miRNAs and adipokines and thus reduce the rate of appearance of GDM and postnatal AGR". The corresponding primary study objective was “to assess whether adipokines/circulating miRNAs levels differed between women from the Mediterranean diet intervention group and those from the control group." However, there was no statistical difference in the incidence of GDM between the control group and the Mediterranean diet intervention group in Table 1. In Table 3, there was no statistically significant difference in Mediterranean diet adherence between participants with and without GDM in the control group and the Mediterranean diet intervention group (both at baseline and 24-28GW). In Table 5, there was no statistically significant difference in fasting blood glucose at 24-28 GW between the control group and the Mediterranean diet intervention group (statistical significance noted by the authors, not statistically significant after reviewer recalculation). Taken together, the results of this manuscript do not support a Mediterranean dietary intervention to prevent the risk of GDM. However, this key is the starting point for this study.
- response The San Carlos GDM prevention study was designed to evaluate whether the effect of a nutritional intervention based on the Mediterranean diet supplemented with EVOO and pistachios, and applied early from the start of pregnancy (before GW 12) could reduce the appearance of GDM. St GDM prevention study demonstrated a 30% reduction in the rate of GDM. This data has been previously reported. For sample size calculation, a primary end-point for conversion to GDM was used assuming an expected conversion rate of 35% between 24-28 GWs and a reduction of at least 30% in the conversion to GDM after intervention. A sample size of 315 women analyzed by group powered 80% to detect a 30% difference between groups in the primary outcomes at 5% significance. This number was increased to 500 per group, to allow for a predicted drop-out of around 20%. Therefore, 1000 women were randomized in the study and we expected to assess at least 700. Finally, a total of 874 women were assessed.
In the current study, for the sample size estimate for the primary objective (miRNAs) a mean in the control group between 8-12 GW of 14.93.10−5 for miR-132-3p, 14.10.10−5 for miR-29a-3p, and 3.09.10−5 for miR-222-3p was expected, according to previous publications. To achieve a relative increase in the mean of at least 20% in each of the miRNAs in the intervention group for a significance level of 5% and a power of 80%, 144 women in each group would be necessary. With this sample size, it would be possible to detect a relative increase in mean adiponectin between the two study groups of more than 12% and a decrease in leptin of 20%, for a significance level of 5% and a power of 80%. (sample size calculation) This sample size does not allow detecting differences in the appearance of GDM between the control and intervention groups. Neither women were selected nor randomized for this objective. Therefore, it is not expected to detect these differences. But these differences have been demonstrated/reported previously (PLoS One. 2017;12(10):e0185873.)
- The statistical methodology of this manuscript needs to be re-examined. For example, IL-6, 8-12GW weight gain, HOMA-IR, fasting insulin, miR-222-3p, miR-103-3p, miR-132, and miR-29a-3p did not seem to conform to a normal distribution. A test for normality should be performed and then the correct statistical method for comparison between groups should be reselected. In addition, there are deviations in the statistical analysis results in many places. Corrections should be recalculated to draw correct conclusions.
3. According to your comments, with which we agree, both insulin/ cytokines/ adipokines and miRNAs do not have a normal distribution and therefore data have been re-evaluated with non-parametric tests. This aspect is included in the statistical study and the levels are expressed in the tables as median (Q1-Q3) instead of Mean + SD. Although the statistical evaluation changes slightly, in our opinion its meaning is maintained. In accordance with these changes, the study of correlations is carried out with the Spearman correlation.
- The authors also put forward the research hypothesis that "circulating levels of aforementioned miRNA could be biomarkers for the appearance of GDM and postnatal AGR, acting through the modification of adipokines and insulin sensitivity". The stated purpose of the study was "The secondary objective is to detect the differences between women who develop GDM and/or postnatal AGR and those who developed neither condition, in an attempt to identify possible early biomarkers of glucose dysregulation". In Table 4 , the miR-222-3p between the GDM group and the control group at 24-28GW was actually not significantly different (statistical significance noted by the authors, not statistically significant after reviewer recalculation). And there was no significant difference in miR-103-3p, miR-132, miR-29a-3p between the two groups. No data were seen for the statistical description of these indicators between the AGR group and the NGT group. In Table 8, the adjusted correlation analysis results show that there is no statistical correlation between the levels of miR-222-3p and miR-103-3p at 24-28GW and 2-3 years of PD and the 2h GTT level of 2-3 years of PD. These results do not support the conclusions (association of miR222-3p with GDM) presented by the authors.
2./3 To find out whether adipokines/cytokines/miRNAs could be biomarkers or characterize women who develop GDM and/or postpartum abnormal glucose regulation, women diagnosed with GDM (GW24) were compared with those who remained normoglycemic during pregnancy (NGT), at baseline (GW12), at the time of GDM screening (24GW) and at postnatal screening (2-3 years postpartum). On the other hand, women who presented abnormal glucose regulation (AGR) at 2 years postpartum were compared with those who were normoglycemic.Likewise, women belonging to the Intervention Group were compared with those of the Control Group with the aim of finding out whether the MedDiet Nutritional Intervention modifies/alters these levels of cytokines/adipokines and miRNA.I believe that our data support the conclusions presented in the paper, which are as follows:
To find out whether there is a metabolic pattern and miRNA associated with GDM, the Group of women with GDM was compared with women with NGT. The results are the following: Women from GDM Group- At baseline (Table 2), in addition to BMI and sBP, (........Women with GDM had a pre-gestational body weight, BMI, sBP values and TNFα levels, median (Q1-Q3), (3.2 (2.0-4.7) vs 2.4 (1.8-3.5) pg/mL; p<0.01) significantly higher than NGT women. Adiponectin levels were lower in the GDM group than in the NGT (15.3 (11.3-24.9)vs.20.0 (14.8-26.4) µg/mL; p=0.008)……) they had a higher TNFα levels and a lower level of Adiponectin: Therefore, only TNFα and Adiponectin could have predictive value, but clearly lower than the predictive value of pre-pregnancy BMI.- At the time of screening for GDM (visit 2, 24-28 GW) and postnatal DM (visit 3, 2-3 years postpartum) (Table 4), GDM is associated with higher levels of leptin and FPI and HOMA-IR values and lower adiponectin levels. In addition, at visit 3 there is a higher level of both IL6 and TNF α and overexpression of miRNA 222.We believe that these data support our conclusions.
To find out whether the nutritional intervention is associated with a certain metabolic pattern, women from the IG are compared with the CG, and always by intention to treat (CG vs IG) and not by protocol (adherence vs low adherence). The results are the following (table 5).- The women of the IG at visit 2 presented a significant decrease in the levels of Insulin, IL6 and Leptin and in the HOMA values, which was associated with an overexpression of miRNA 103 and 222. This persists at visit 3 except for Leptin, which loses its statistically significance. Beyond speculation. I believe that the data provided support our conclusions
